# Personalized Privacy Amplification via Importance Sampling

**Dominik Fay**                                                                    *dominikf@kth.se*
*Elekta and KTH Royal Institute of Technology, Stockholm, Sweden*

**Sebastian Mair**                                                          *sebastian.mair@liu.se*
*Linköping University and Uppsala University, Sweden*

**Jens Sjölund**                                                            *jens.sjolund@it.uu.se*
*Uppsala University, Sweden*

**Reviewed on OpenReview:** *https://openreview.net/forum?id=IK2cR89z45*

## Abstract

For scalable machine learning on large data sets, subsampling a representative subset is a common approach for efficient model training. This is often achieved through importance sampling, whereby informative data points are sampled more frequently. In this paper, we examine the privacy properties of importance sampling, focusing on an individualized privacy analysis. We find that, in importance sampling, privacy is well aligned with utility but at odds with sample size. Based on this insight, we propose two approaches for constructing sampling distributions: one that optimizes the privacy-efficiency trade-off; and one based on a utility guarantee in the form of coresets. We evaluate both approaches empirically in terms of privacy, efficiency, and accuracy on the differentially private $k$-means problem. We observe that both approaches yield similar outcomes and consistently outperform uniform sampling across a wide range of data sets. Our code is available on GitHub.[1]

## 1 Introduction

When deploying machine learning models in practice, two central challenges are scalability, *i.e.*, the computationally efficient handling of large data sets, and the protection of user privacy. A common approach to the former challenge is subsampling, *i.e.*, performing demanding computations only on a subset of the data, see, *e.g.*, Alain et al. (2016); Katharopoulos & Fleuret (2018). Here, importance sampling is a powerful tool that can reduce the variance of the subsampled estimator. It assigns higher sampling probabilities to data points that are more informative for the task at hand while keeping the estimate unbiased. For instance, importance sampling is commonly used to construct coresets, that is, subsets whose loss function value is provably close to the loss for the full data set, see, *e.g.*, Bachem et al. (2018). For the latter challenge, differential privacy (Dwork et al., 2006b) offers a framework for publishing trained models in a way that respects the individual privacy of every user.

Differential privacy and subsampling are related via the concept of *privacy amplification by subsampling* (Kasiviswanathan et al., 2008; Balle et al., 2018), which states, loosely speaking, that subsampling with probability $p$ improves the privacy parameter of a subsequently run differentially private algorithm by a factor of approximately $p$. A typical application of this result involves re-scaling the query by a factor of $1/p$ to eliminate the sampling bias, thereby approximately canceling out the privacy gains, but keeping the efficiency gain. It forms the foundation for many practical applications of differential privacy, such as differentially private stochastic gradient descent (Bassily et al., 2014; Abadi et al., 2016).

So far, privacy amplification has been predominantly used with uniform sampling (Steinke, 2022). Although the potential of data-dependent sampling for reducing sampling variance is well understood (*e.g.*, Robert

---

[1] https://github.com/smair/personalized-privacy-amplification-via-importance-sampling

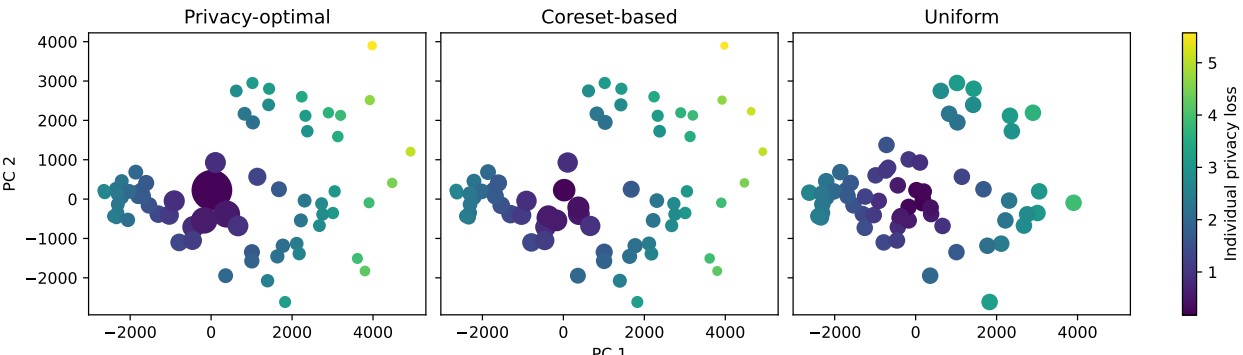

Figure 1: Illustration of three subsampling strategies for the learning task of $k$-means clustering on the *Song* data set. We show a scatter plot of the first two principal components of the sampled points. The marker size is proportional to the importance weight, while the color represents the individual privacy loss before sampling. **Left**: Our privacy-constrained sampling selects data points with higher individual privacy loss more frequently and with lower weight. **Middle**: Coreset-based sampling selects data points based on their potential impact on the objective function. **Right**: Uniform sampling selects data points with equal probability.

& Casella (2005)), it has largely remained untapped in differential privacy because its privacy benefits have not been as clear (Bun et al., 2022; Drechsler & Bailie, 2024). A longstanding objection to data-dependent sampling is that the privacy amplification factor scales with the maximum sampling probability when applied to heterogeneous probabilities, leading to worse privacy than uniform sampling when controlling for sample size. Recently, Bun et al. (2022) confirmed that this also holds for probability-proportional-to-size sampling – a sampling strategy closely related to importance sampling – and further noted that additional privacy leakage may arise from data points influencing other data points' sampling probabilities.

We find that these challenges can be addressed with an appropriate sampling scheme and an individualized privacy analysis incorporating more information than previous work. We propose Poisson importance sampling, defined as sampling each data point independently with a probability that depends only on the data point itself and weighting the point by the reciprocal of the probability. In this setting, we conduct a privacy analysis that explicitly considers the individual privacy loss of each data point when sampled with a specific weight and probability. Our analysis reveals that, in importance sampling, privacy is typically not at odds with utility but with *sample size*. This is for two reasons: **(i)** the most informative points typically have the highest privacy loss, and **(ii)** when importance weights are accounted for, decreasing the sampling probability typically *increases* the privacy loss. Perhaps counterintuitively, we conclude that we should assign high sampling probabilities to data points with high privacy loss to have good privacy *and* good utility.

At first, statement **(ii)** might seem to suggest that we should reject weighted sampling altogether because there is no privacy gain. However, this is misleading because the statement applies even more to the widely used uniform sampling when weights are accounted for. Indeed, we find that importance sampling is superior to uniform sampling in terms of mitigating the impact on privacy and utility. By including sampling weights into the framework of privacy amplification, we make this effect explicit and highlight that the primary purpose of subsampling (whether uniform or data-dependent) is to improve the efficiency of the mechanism, not its privacy-utility trade-off.

Based on this insight, we derive two specific approaches for constructing importance sampling distributions for a given mechanism.

- Our first approach navigates the aforementioned adverse relationship between privacy and sample size by minimizing the expected sample size subject to a worst-case privacy constraint. This approach effectively equalizes the individual privacy losses of the mechanism. We call this approach *privacy-constrained* sampling. It applies to any differentially private mechanism whose individual privacy

loss profile is known. We provide an efficient algorithm that optimizes the sampling probabilities numerically.

- Our second approach is based on the concept of coresets, which are small, representative subsets that come with strong utility guarantees in the form of a confidence interval around the loss function value of the full data set. We derive the privacy properties of a coreset-based sampling distribution when used in conjunction with differentially private $k$-means clustering.

The two approaches are visualized in Figure 1 alongside uniform sampling. The figure shows that the privacy-constrained weights are highly correlated to the utility-based coreset weights, supporting the intuition that privacy and utility are well aligned in importance sampling.

We empirically evaluate the proposed approaches on the task of differentially private $k$-means clustering. We compare them to uniform sampling in terms of efficiency, privacy, and accuracy on eight different data sets of varying sizes (cf. Table 1). We find that both of our approaches consistently provide better utility for a given sample size and privacy budget than uniform sampling on all data sets. We find meaningful privacy-utility improvements even in the medium-to-low privacy regime where uniform sampling typically fails to do so. One practical implication is that importance sampling can be used effectively to subsample the data set *once* at the beginning of the computation, while uniform subsampling typically requires repeated subsampling at each iteration.

## 2 Preliminaries

We begin by stating the necessary concepts that also introduce the notation used in this paper. We denote by $\mathcal{X} \subseteq \mathbb{R}^d$ the set of all possible data points. A data set $\mathcal{D} \in \mathcal{X}^*$ is a finite subset of $\mathcal{X}$. We use $\mathcal{B}_{d,p}(r) = \{\mathbf{x} \in \mathbb{R}^d \mid \|\mathbf{x}\|_p \leq r\}$ to refer to the $\ell_p$-norm ball of radius $r$ in $d$ dimensions. When the dimension is clear from context, we omit the subscript $d$ and write $\mathcal{B}_p(r)$.

**Differential Privacy.** Differential privacy (DP) is a formal notion of privacy stating that the output of a data processing method should be robust, in a probabilistic sense, to changes in the data set that affect individual data points. It was introduced by Dwork et al. (2006b) and has since become a standard tool for privacy-preserving data analysis (Ji et al., 2014). The notion of robustness is qualified by a parameter $\epsilon \geq 0$ that relates to the error rate of any adversary that tries to infer whether a particular data point is present in the data set (Kairouz et al., 2015; Dong et al., 2019). Differential privacy is based on the formal notion of indistinguishability.

**Definition 1** ($\epsilon$-indistinguishability). Let $\epsilon \geq 0$. Two distributions $P, Q$ on a space $\mathcal{Y}$ are $\epsilon$-*indistinguishable* if $P(Y) \leq e^\epsilon Q(Y)$ and $Q(Y) \leq e^\epsilon P(Y)$ for all measurable $Y \subseteq \mathcal{Y}$.

**Definition 2** ($\epsilon$-DP). A randomized mechanism $\mathcal{M} \colon \mathcal{X}^* \to \mathcal{Y}$ is said to be $\epsilon$-*differentially private* ($\epsilon$-DP) if, for all neighboring data sets $\mathcal{D}, \mathcal{D}' \in \mathcal{X}^*$, the distributions of $\mathcal{M}(\mathcal{D})$ and $\mathcal{M}(\mathcal{D}')$ are $\epsilon$-indistinguishable. Two data sets $\mathcal{D}, \mathcal{D}'$ are neighboring if $\mathcal{D} = \mathcal{D}' \cup \{\mathbf{x}\}$ or $\mathcal{D}' = \mathcal{D} \cup \{\mathbf{x}\}$ for some $\mathbf{x} \in \mathcal{X}$.

Formally, we consider a randomized mechanism as a mapping from a data set to a random variable. Differential privacy satisfies several convenient analytical properties, including closedness under post-processing (Dwork et al., 2006b), composition (Dwork et al., 2006b;a; 2010; Kairouz et al., 2015), and sampling (Kasiviswanathan et al., 2008; Balle et al., 2018). The latter is called *privacy amplification by subsampling*.

**Proposition 3** (Privacy Amplification by Subsampling). *Let $\mathcal{D}$ be a data set of size $n$, $\mathcal{S}$ be a subset of $\mathcal{D}$ where every $\mathbf{x}$ has a constant probability $p$ of being independently sampled, i.e., $q(\mathbf{x}) = p \in (0, 1]$, and $\mathcal{M}$ be an $\epsilon$-DP mechanism. Then, $\mathcal{M}(\mathcal{S})$ satisfies $\epsilon'$-DP for $\epsilon' = \log(1 + (\exp(\epsilon) - 1)p)$.*

In the case of heterogeneous sampling probabilities $p_1, \ldots, p_n$, the privacy amplification scales with $\max_i p_i$ instead of $p$. It is important to note that this only provides meaningful privacy amplification if $\epsilon$ is sufficiently small. For $\epsilon > 1$, we only have $\epsilon' \approx \epsilon - \log(1/p)$.

**Personalized Differential Privacy.** In many applications, the privacy loss of a mechanism is not uniform across the data set. This heterogeneity can be captured by the notion of personalized differential privacy (PDP) (Jorgensen et al., 2015; Ebadi et al., 2015; Alaggan et al., 2016).

**Definition 4** (Personalized differential privacy). *Let $\epsilon\colon \mathcal{X} \to \mathbb{R}_{\geq 0}$. A mechanism $\mathcal{M} : \mathcal{X}^* \to \mathcal{Y}$ is said to satisfy $\epsilon$-personalized differential privacy ($\epsilon$-PDP) if, for all data sets $\mathcal{D}$ and differing points $\mathbf{x} \in \mathcal{X}$, the distributions of $\mathcal{M}(\mathcal{D})$ and $\mathcal{M}(\mathcal{D} \cup \{\mathbf{x}\})$ are $\epsilon(\mathbf{x})$-indistinguishable. We call the function $\epsilon(\cdot)$ a PDP profile of $\mathcal{M}$.*

**Importance Sampling.** In learning problems, the (weighted) objective function that we optimize is often a sum over per-point losses, *i.e.*, $\phi_{\mathcal{D}} = \sum_{\mathbf{x} \in \mathcal{D}} w(\mathbf{x}) \ell(\mathbf{x})$. Here, we typically have $w(\mathbf{x}) = 1 \; \forall \mathbf{x} \in \mathcal{D}$ for the full data $\mathcal{D}$. Uniformly subsampling a data set yields an unbiased estimation of the mean. A common way to improve upon uniform sampling is to skew the sampling towards important data points while retaining an unbiased estimate of the objective function. Let $q$ be an importance sampling distribution on $\mathcal{D}$ which we use to sample a subset $\mathcal{S}$ of $m \ll n$ points and weight them with $w(\mathbf{x}) = (mq(\mathbf{x}))^{-1}$. Then, the estimation of the objective function is unbiased, *i.e.*, $\mathbb{E}_{\mathcal{S}}[\phi_{\mathcal{S}}] = \phi_{\mathcal{D}}$.

# 3 Privacy Amplification via Importance Sampling

This section presents our framework for importance sampling with differential privacy. In Section 3.1, we introduce the notion of Poisson importance sampling and state and discuss its general privacy properties. Section 3.2 presents our first approach for deriving importance sampling distributions, namely privacy-constrained sampling. Finally, in Section 3.3, we give a numerical example that illustrates the aforementioned results and compares them to uniform subsampling for the Laplace mechanism. We defer all proofs to Appendix D.

## 3.1 General Sampling Distributions

We begin by introducing *Poisson importance sampling*, which is the sampling strategy we use throughout the paper. It is a weighted version of the Poisson sampling strategy described in Proposition 3.

**Definition 5** (Poisson Importance Sampling). *Let $q\colon \mathcal{X} \to [0,1]$ be a function, and $\mathcal{D} = \{\mathbf{x}_1, \ldots, \mathbf{x}_n\} \subseteq \mathcal{X}$ be a data set. A Poisson importance sampler for $q$ is a randomized mechanism $S_q(\mathcal{D}) = \{(w_i, \mathbf{x}_i) \mid \gamma_i = 1\}$, where $w_i = 1/q(\mathbf{x}_i)$ are weights and $\gamma_1, \ldots, \gamma_n$ are independent Bernoulli variables with parameters $p_i = q(\mathbf{x}_i)$.*

By having each probability $q(\mathbf{x}_i)$ depend only on the data point $\mathbf{x}_i$ itself and keeping the selection events independent, we ensure that the influence of any single data point on the sample is small. It is important to note that the function definition of $q$ must be data-independent and considered public information, *i.e.*, fixed before observing the data set, while the specific probabilities $\{q(\mathbf{x}_i)\}_{i=1}^n$ evaluated on the data set are not published. Note also that Poisson importance sampling outputs a weighted data set. We intend for the subsequent mechanism to use these weights to offset the sampling bias, *e.g.*, by using a weighted sum when the goal is to estimate a population sum. However, note that our formal privacy results also apply to biased mechanisms.

In order to characterize the privacy properties of Poisson importance sampling, we analyze the impact of the sampling probability jointly with the data point's individual privacy loss in the base mechanism. For this reason, we express our results within PDP, which is in contrast to previous characterizations of data-dependent sampling (*e.g.*, Bun et al. (2022)). Our first result describes the general case of an arbitrary PDP mechanism subsampled with an arbitrary importance sampling distribution.

**Theorem 6** (Amplification by Importance Sampling). *Let $\mathcal{M} : [1, \infty) \times \mathcal{X}^* \to \mathcal{Y}$ be an $\epsilon$-PDP mechanism that operates on weighted data sets, $q\colon \mathcal{X} \to [0,1]$ be a function, and $S_q(\cdot)$ be a Poisson importance sampler for $q$. The mechanism $\widehat{\mathcal{M}} = \mathcal{M} \circ S_q$ satisfies $\psi$-PDP with*

$$\psi(\mathbf{x}) = \log\Big(1 + q(\mathbf{x})\big(e^{\epsilon(w, \mathbf{x})} - 1\big)\Big), \quad \text{where} \quad w = \frac{1}{q(\mathbf{x})}. \tag{1}$$

The sampled PDP profile in Equation (1) closely resembles the privacy amplification result for uniform sampling (Proposition 3) with the important distinction that $\epsilon(w, \mathbf{x})$ depends on a weight $w$ and a data point $\mathbf{x}$. This relatively small change has important implications for the trade-offs between privacy, efficiency, and accuracy. In general, it is no longer obvious whether the privacy loss increases or decreases as a function of the sampling probability. However, we show in Appendix A that, for an important class of PDP profiles, the sampled PDP profile is decreasing in $q(\mathbf{x})$. This class is important because it includes PDP profiles that

are linear in $w$. In the context of importance sampling, we argue that all mechanisms of interest are linear in $w$, because linearity follows from a simple invariance condition: we should expect a mechanism to treat a weighted data point $(w, \mathbf{x})$ the same as if it were $w$ distinct data points of value $\mathbf{x}$, each with weight 1. For (generalized) linear queries, this invariance coincides with unbiasedness and thus agrees with the intuitive purpose of an importance weight. If this invariance holds, the PDP profile is linear in $w$ due to group privacy.

From the above discussion, we conclude that a good sampling distribution should assign high probabilities to informative data points in order to achieve good privacy and utility. Hence, the privacy-utility-efficiency trilemma reduces to a dilemma between efficiency on the one hand; and privacy and utility on the other. This suggests two natural approaches to constructing sampling distributions: one that optimizes the privacy-efficiency trade-off and one that optimizes the utility-efficiency trade-off. We explore the first approach in Section 3.2 and the second approach in Section 4.

### 3.2 Sampling with optimal privacy-efficiency trade-off

We now describe how to construct a sampling distribution that achieves a given $\epsilon$-DP guarantee with minimal expected sample size. The motivation for this is twofold. First, as we have seen in the previous subsection, sample size is the primary limiting factor to privacy in importance sampling and additionally serves as the primary indicator of efficiency. Secondly, by imposing a constant PDP bound as a constraint we can ensure that the subsampled mechanism satisfies $\epsilon$-DP by design. This is not obvious from Theorem 6, as the resulting PDP profile may be unbounded.

Minimizing the expected sample size subject to a given $\epsilon^\star$-DP constraint can be described as the following optimization problem.

**Problem 7** (Privacy-constrained sampling). *For a PDP profile $\epsilon\colon [1, \infty) \times \mathcal{X} \to \mathbb{R}_{\geq 0}$, a target privacy guarantee $\epsilon^\star$, and a data set $\mathcal{D} \subseteq \mathcal{X}$ of size $n$, we define the optimization problem for privacy-constrained sampling as*

$$\underset{w_1, \ldots, w_n}{\text{minimize}} \quad \sum_{i=1}^{n} \frac{1}{w_i} \tag{2a}$$

$$\text{subject to} \quad \log\left(1 + \frac{1}{w_i}\left(e^{\epsilon(w_i, \mathbf{x}_i)} - 1\right)\right) \leq \epsilon^\star \qquad \text{for all } i, \tag{2b}$$

$$w_i \geq 1, \qquad\qquad\qquad\qquad \text{for all } i. \tag{2c}$$

The constraint in Equation (2b) captures the requirement that $\psi$ should be bounded by $\epsilon^\star$ for all $\mathbf{x} \in \mathcal{X}$, and the constraint in Equation (2c) ensures that $1/w_i$ is a probability. When the PDP profile is linear in $w$, the problem can be solved with standard convex optimization techniques. Below, we provide a more general result that guarantees a unique solution and an efficient algorithm for a broad class of (possibly) nonconvex PDP profiles. In order to guarantee a unique solution, we require the following mild regularity conditions.

**Assumption 8.** For all $\mathbf{x} \in \mathcal{X}$, $\epsilon^\star \geq \epsilon(1, \mathbf{x})$.

**Assumption 9.** For all $\mathbf{x} \in \mathcal{X}$, there is a constant $v_\mathbf{x} \geq 1$ such that $\epsilon(w, \mathbf{x}) > \log(1 + w(e^{\epsilon^\star} - 1))$ for all $w \geq v_\mathbf{x}$.

Assumption 8 ensures that the feasible region is non-empty, because $w_1 = w_2 = \cdots = w_n = 1$ always satisfies the constraints, while Assumption 9 essentially states that, asymptotically, $\epsilon$ should grow at least logarithmically fast with $w$, ensuring that the feasible region is bounded. Formally, our existence result is as follows.

**Theorem 10.** *Let Assumptions 8 and 9 be satisfied. There is a data set-independent function $w\colon \mathcal{X} \to [1, \infty)$, such that, for all data sets $\mathcal{D} \in \mathcal{X}^*$, Problem 7 has a unique solution $w^\star(\mathcal{D}) = (w_1^\star(\mathcal{D}), \ldots, w_n^\star(\mathcal{D}))$ of the form $w_i^\star(\mathcal{D}) = w(\mathbf{x}_i)$. Furthermore, let $\mathcal{M}$ be a mechanism that admits the PDP profile $\epsilon$ and $S_q$ be a Poisson importance sampler for $q(\mathbf{x}) = 1/w(\mathbf{x})$. Then, $\mathcal{M} \circ S_q$ satisfies $\epsilon^\star$-DP.*

The fact that the weights are of the form $w_i^*(\mathcal{D}) = w(\mathbf{x}_i)$ is important for privacy. It ensures that the probability map $q(\cdot)$ is only a function of $\mathbf{x}_i$ and not of the remainder of the data set, which is a requirement for Poisson importance sampling.

Note also that, with privacy-constrained sampling, we can achieve some sampling "for free": if the base mechanism satisfies $\epsilon$-DP and we choose $\epsilon^* = \epsilon$, then the sampled mechanism also satisfies $\epsilon$-DP while operating on a smaller sample.

Next, in Algorithm 1, we describe an efficient algorithm to solve privacy-constrained sampling. We reduce Problem 7 to a scalar root-finding problem and solve it via bisection. Since the sampled PDP profile might have many roots, we require an additional assumption in order to identify which root is optimal. We introduce Assumption 11 which is slightly stronger than Assumption 9, but still permits a variety of nonconvex PDP profiles.

**Assumption 11.** The function $\epsilon(w, \mathbf{x})$ is differentiable w.r.t. $w$ and $\exp \circ \epsilon$ is $\mu_{\mathbf{x}}$-strongly convex in $w$ for all $\mathbf{x}$.

---

**Algorithm 1** Optimization for privacy-constrained importance weights

---

1: **Input:** Data set $\mathcal{D} = \{\mathbf{x}_1, \ldots, \mathbf{x}_n\}$, target privacy guarantee $\epsilon^* > 0$, PDP profile $\epsilon$, its derivative $\epsilon'$ with respect to $w$, and strong convexity constants $\mu_1, \ldots, \mu_n$
2: **Output:** Importance weights $w_1, \ldots, w_n$
3: **for** $i = 1, \ldots, n$ **do**
4:    Define $g_i(w) = \frac{1}{w}\left(e^{\epsilon(w, \mathbf{x}_i)} - 1\right) - \left(e^{\epsilon^*} - 1\right)$
5:    $\bar{v}_i \leftarrow \min\{e^{\epsilon(1, \mathbf{x}_i)} + \mu_i/2, \ \epsilon'(1, \mathbf{x}_i)e^{\epsilon(1, \mathbf{x}_i)} + 1\}$
6:    $b_i \leftarrow 2(e^{\epsilon^*} - \bar{v}_i)/\mu_i + 1$
7:    **if** $\epsilon(1, \mathbf{x}_i) = \epsilon^*$ and $\epsilon'(1, \mathbf{x}_i) < 0$ **then**
8:        $w_i \leftarrow$ Bisect $g_i$ with initial bracket $(1, b_i]$
9:    **else**
10:       $w_i \leftarrow$ Bisect $g_i$ with initial bracket $[1, b_i]$
11:   **end if**
12: **end for**

---

As the following proposition shows, Algorithm 1 solves Problem 7 using only a few evaluations of $\epsilon(w, \mathbf{x})$. We note that faster solutions are possible, *e.g.*, via Newton's method, but we have found bisection to be sufficiently fast in our experiments (see Section 5).

**Proposition 12.** *Let Assumptions 8 and 11 be satisfied. Algorithm 1 solves Problem 7 up to accuracy $\alpha$ with at most $\sum_{\mathbf{x} \in \mathcal{D}} \log_2\lceil (e^{\epsilon^*} - \bar{v}_{\mathbf{x}})/(\alpha \mu_{\mathbf{x}})\rceil$ evaluations of $\epsilon(w, \mathbf{x})$, where $\epsilon'(w, \mathbf{x}) = \partial/\partial w\, \epsilon(w, \mathbf{x})$ and $\bar{v}_{\mathbf{x}} = \min\{e^{\epsilon(1, \mathbf{x})} + \mu_{\mathbf{x}}/2, \ \epsilon'(1, \mathbf{x})e^{\epsilon(1, \mathbf{x})} + 1\}$.*

### 3.3 Importance sampling for the Laplace mechanism

We conclude this section with a simple numerical example for Theorems 6 and 10 to illustrate that (i) privacy and utility are well-aligned goals in importance sampling and (ii) uniform sampling is highly suboptimal even for very fundamental mechanisms. For this purpose, we generate synthetic data on which we run a Laplacian sum mechanism and compare privacy-constrained sampling to uniform sampling as well as to an idealized benchmark in terms of privacy and variance at a fixed expected sample size.

Let $\mathcal{D} = \{\mathbf{x}_i\}_{i=1}^n$ be a data set and $\mathcal{I} \subseteq \{1, \ldots, n\}$ a subset of indices. We consider the Laplacian weighted sum mechanism $\mathcal{M}_{\mathrm{LWS}}(\widetilde{\mathcal{D}}) = \zeta + \sum_{i \in \mathcal{I}} w_i \mathbf{x}_i$ as an example, where $\widetilde{\mathcal{D}} = \{(w_i, \mathbf{x}_i)\}_{i \in \mathcal{I}}$ is a weighted subset of $\mathcal{D}$ and $\zeta$ is standard Laplace distributed. The PDP profile of $\mathcal{M}_{\mathrm{LWS}}$ is given by $\epsilon_{\mathrm{LWS}}(w, \mathbf{x}) = w\|\mathbf{x}\|_1$. When $\widetilde{\mathcal{D}}$ is obtained by Poisson importance sampling $\mathcal{S}_q$ from $\mathcal{D}$, then the mechanism $\mathcal{M}_{\mathrm{LWS}} \circ \mathcal{S}_q$ has variance

$$\mathrm{Var}\left[\mathcal{M}_{\mathrm{LWS}}(\widetilde{\mathcal{D}})_j\right] = 2 + \sum_{i=1}^n \left(\frac{1}{q(\mathbf{x}_i)} - 1\right)[\mathbf{x}_i]_j^2$$

in the $j$-th dimension, where the randomness is over both the noise and the sampling. We compare three sampling strategies: uniform sampling $q_{\mathrm{unif}}(\mathbf{x}) = m/n$, privacy-constrained sampling $q_{\mathrm{priv}}$ according to Theorem 10, and an idealized benchmark which we call variance-optimal sampling $q_{\mathrm{var}}$, defined as the

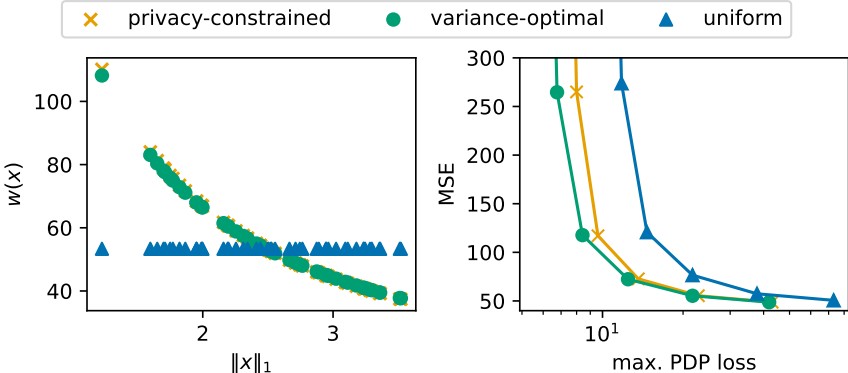

Figure 2: Comparison of sampling strategies for the Laplace sum mechanism. **Left**: The importance weight for each data point. Each marker in the plot represents one data point. The points are ordered by their $\ell_1$-norm. **Right**: Estimation error for varying noise scales. Lower is better. The horizontal axis is the maximum PDP loss over the data set.

solution to the following optimization problem:

$$\underset{q}{\text{minimize}} \quad \sum_{j=1}^{d} \text{Var}\left[\mathcal{M}_{\text{LWS}}(\widetilde{\mathcal{D}})_j\right] \quad \text{subject to} \quad \sum_{i=1}^{n} q(\mathbf{x}_i) = m.$$

The variance-optimal distribution serves as a lower bound on the variance achievable by any DP sampling distribution. It does not satisfy $\epsilon$-DP itself, as it requires oracle access to the data set. As we show below, the privacy-constrained distribution achieves performance close to variance-optimal while also satisfying $\epsilon$-DP.

We generate $n = 1000$ points from an isotropic multivariate normal distribution in $d = 10$ dimensions with variance $\sigma^2 = 1/d$ in each dimension. First, we visualize the importance weights $w(\mathbf{x}_i)$ for each sampling strategy. For this, we fix a target sample size at $m = 19$ and compute the weights $w(\mathbf{x}_i)$ for each sampling strategy that achieve the target sample size in expectation. The results are shown in Figure 2 (left). Remarkably, the privacy-constrained weights and the variance-optimal weights are almost identical.

Next, we compare mean-squared error (MSE) and privacy loss of the sampling strategies at different privacy levels. For this, we vary the noise scale $b$ of the Laplace mechanism over a coarse grid in $[3, 3000]$. For each $b$, we fix the expected sample size $m_b$ by computing the privacy-constrained weights. Then, we compute the corresponding weights for the other two sampling strategies such that they achieve $m_b$ in expectation. We then compute the individual PDP losses $\psi(\mathbf{x}_i)$ for each sampling strategy and obtain their respective maximum over the data set. Finally, we compute the MSE between the quantities $\sum_{\mathbf{x}_i \in \mathcal{D}} \mathbf{x}_i$ and $\mathcal{M}_{\text{LWS}}(\tilde{\mathcal{D}})$ of each sampling strategy by averaging over 1000 independent runs. The resulting PDP losses and MSEs in Figure 2 (right) show substantial improvements of privacy-constrained sampling over uniform sampling.

## 4 Importance Sampling for DP $k$-means

In this section, we apply our results on privacy amplification for importance sampling to the differentially private $k$-means problem. We define a weighted version of the DP-Lloyd algorithm, and analyze its PDP profile. Then, we derive the required constants for the privacy-constrained distribution and establish a privacy guarantee for a coreset-based sampling distribution. All proofs are deferred to Appendix D.

**Differentially Private $k$-means Clustering.** Given a data set $\mathcal{D}$ as introduced before, the goal of $k$-means clustering is to find a set of $k \in \mathbb{N}$ cluster centers $\mathcal{C} = \{\mathbf{c}_1, \ldots, \mathbf{c}_k\}$ that minimize the distance of the data points to their closest cluster center. The objective function is given as $\phi_{\mathcal{D}}(\mathcal{C}) = \sum_{i=1}^{n} d(\mathbf{x}_i, \mathcal{C})$, where $d(\cdot, \cdot)$ is the distance of a point $\mathbf{x}$ to its closest cluster center $\mathbf{c}_j$, which is $d(\mathbf{x}, \mathcal{C}) = \min_j \|\mathbf{x} - \mathbf{c}_j\|_2^2$. The standard approach of solving the $k$-means problem is Lloyd's algorithm (Lloyd, 1982). It starts by initializing the cluster

centers $\mathcal{C}$ and then alternates between two steps. First, given the current cluster centers $\mathcal{C}$, it computes the cluster assignments $\mathcal{C}_j = \{\mathbf{x}_i \mid j = \arg\min_j \|\mathbf{x}_i - \mathbf{c}_j\|_2^2\}$ for every $j = 1, \ldots, k$. Second, it updates all cluster centers to be the mean of their cluster $\mathbf{c}_j = \frac{1}{|\mathcal{C}_j|} \sum_{\mathbf{x} \in \mathcal{C}_j} \mathbf{x}$. Those two steps are iterated until convergence.

The standard way to make Lloyd's algorithm differentially private is to add appropriately scaled noise to both steps of the algorithm. Gaussian noise has been suggested (Blum et al., 2005), resulting in $(\epsilon, \delta)$-DP and Laplace noise has been suggested (Su et al., 2016; 2017) for $\ell_1$ geometry, resulting in $\epsilon$-DP. Here, we give a generalized version based on the exponential mechanism (McSherry & Talwar, 2007) that guarantees $\epsilon$-DP for any $\ell_p$ geometry. We refer to it as DP-Lloyd. Let $\xi \in \mathbb{R}^k$ be a random vector whose entries independently follow a zero mean Laplace distribution with scale $\beta_{\text{count}} > 0$. Furthermore, let $\zeta_1, \ldots, \zeta_k \in \mathbb{R}^d$ be independent random vectors, each drawn from the density $p(\zeta) \propto \exp(-\|\zeta\|_p/\beta_{\text{sum}})$. The cluster centers are then updated as follows:

$$\mathbf{c}_j = \frac{1}{\xi_j + |\mathcal{C}_j|}\left(\zeta_j + \sum_{\mathbf{x} \in \mathcal{C}_j} \mathbf{x}\right) \quad \text{for } j = 1, 2, \ldots, k.$$

Assuming the points have bounded $\ell_p$-norm, i.e., $\mathcal{X} = \mathcal{B}_p(r)$ for some $r > 0$, then DP-Lloyd preserves $\epsilon$-DP with $\epsilon = (r/\beta_{\text{sum}} + 1/\beta_{\text{count}})T$ after $T$ iterations.

**Weighted DP Lloyd's Algorithm.** In order to apply importance subsampling to $k$-means, we first define a weighted version of DP-Lloyd. Each iteration of DP-Lloyd consists of a counting query and a sum query, which generalize naturally to the weighted scenario. Specifically, for a weighted data set $\mathcal{S} = \{(w_i, \mathbf{x}_i)\}_{i=1}^n$, we define the update step to be

$$\mathbf{c}_j = \frac{1}{\xi_j + \sum_{(w,\mathbf{x}) \in \mathcal{C}_j} w}\left(\zeta_j + \sum_{(w,\mathbf{x}) \in \mathcal{C}_j} w\mathbf{x}\right) \quad \text{for } j = 1, 2, \ldots, k.$$

We summarize this in Algorithm 2 which can be found in Appendix D. This approach admits the following PDP profile which generalizes naturally from the $\epsilon$-DP guarantee of DP-Lloyd.

**Proposition 13.** *The weighted DP Lloyd algorithm (Algorithm 2) satisfies the PDP profile*

$$\epsilon_{\text{Lloyd}}(w, \mathbf{x}) = \left(\frac{1}{\beta_{\text{count}}} + \frac{\|\mathbf{x}\|_p}{\beta_{\text{sum}}}\right)Tw. \tag{3}$$

**Privacy-constrained sampling.** In order to compute the privacy-constrained weights via Algorithm 1, we need the strong convexity constant of $\epsilon_{\text{Lloyd}}$, which is readily obtained via the second derivative:

$$\frac{\partial^2}{\partial^2 w}\exp(\epsilon_{\text{Lloyd}}(w, \mathbf{x})) \geq T^2\left(\frac{1}{\beta_{\text{count}}} + \frac{\|\mathbf{x}\|_p}{\beta_{\text{sum}}}\right)\exp\left(\left(\frac{1}{\beta_{\text{count}}} + \frac{\|\mathbf{x}\|_p}{\beta_{\text{sum}}}\right)T\right) =: \mu. \tag{4}$$

Besides the privacy-constrained distribution, we also consider a coreset-based sampling distribution. Before doing so, we first introduce the idea of a coreset.

**Coresets for $k$-means.** A coreset is a weighted subset $\mathcal{S} \subseteq \mathcal{D}$ of the full data set $\mathcal{D}$ with cardinality $m \ll n$, on which a model performs provably competitive when compared to the performance of the model on $\mathcal{D}$. Since we are now dealing with weighted data sets, we define the weighted objective of $k$-means to be $\phi_{\mathcal{D}}(\mathcal{C}) = \sum_{\mathbf{x} \in \mathcal{D}} w(\mathbf{x})d(\mathbf{x}, \mathcal{C})$, where $w(\mathbf{x}) \geq 0$ are the non-negative weights. In this paper, we use a sampling distribution inspired by a lightweight coreset construction as introduced by Bachem et al. (2018).

**Definition 14** (Lightweight coreset). *Let $\varepsilon > 0$, $k \in \mathbb{N}$, and $\mathcal{D} \in \mathcal{X}^*$ be a set of points with mean $\bar{\mathbf{x}}$. A weighted set $\mathcal{S}$ is a $(\varepsilon, k)$-lightweight coreset of the data $\mathcal{D}$ if for any $\mathcal{C} \subseteq \mathbb{R}^d$ of cardinality at most $k$ we have $|\phi_{\mathcal{D}}(\mathcal{C}) - \phi_{\mathcal{S}}(\mathcal{C})| \leq \frac{\varepsilon}{2}\phi_{\mathcal{D}}(\mathcal{C}) + \frac{\varepsilon}{2}\phi_{\mathcal{D}}(\{\bar{\mathbf{x}}\}).$*

Note that the definition holds for any choice of cluster centers $\mathcal{C}$. Bachem et al. (2018) propose the sampling distribution $q(\mathbf{x}) = \frac{1}{2}\frac{1}{n} + \frac{1}{2}\frac{d(\mathbf{x}, \bar{\mathbf{x}})}{\sum_{i=1}^n d(\mathbf{x}_i, \bar{\mathbf{x}})}$ and assign each sampled point $\mathbf{x}$ the weight $(mq(\mathbf{x}))^{-1}$. This is a mixture distribution of a uniform and a data-dependent part. Note that by using these weights, $\phi_{\mathcal{S}}(\mathcal{C})$ yields an unbiased estimator of $\phi_{\mathcal{D}}(\mathcal{C})$. The following theorem shows that this sampling distribution yields a lightweight coreset with high probability.

**Theorem 15** (Bachem et al. (2018)). *Let $\varepsilon > 0$, $\Delta \in (0,1)$, $k \in \mathbb{N}$, $\mathcal{D}$ be a set of points in $\mathcal{X}$, and $\mathcal{S}$ be the sampled subset according to $q$ with a sample size $m$ of at least $m \geq c\frac{kd\log k - \log \Delta}{\varepsilon^2}$, where $c > 0$ is a constant. Then, with probability of at least $1 - \Delta$, $\mathcal{S}$ is an $(\varepsilon, k)$-lightweight-coreset of $\mathcal{D}$.*

**Coreset-based sampling distribution.** We adapt the sampling distribution from Theorem 15 to the Poisson sampling setting and propose $q(\mathbf{x}) = \lambda\frac{m}{n} + (1-\lambda)\frac{m\|\mathbf{x}\|_2^2}{n\tilde{\mathbf{x}}}$, where $m \ll n$ is the expected subsample size and $\tilde{\mathbf{x}} = \frac{1}{n}\sum_{i=1}^{n}\|\mathbf{x}_i\|_2^2$ is the average squared $\ell_2$-norm. To ensure proper probabilities, it is necessary to constrain the subsampling size to $m \leq n\tilde{x}r^{-2}$. Compared to Bachem et al. (2018), there are three changes: (i) the change to a Poisson sampling setting, (ii) the assumption that the data set $\mathcal{X}$ is centered, and (iii) the introduction of $\lambda \in [0,1]$ which yields a uniform sampler for the choice of $\lambda = 1$.

We compute the $\epsilon$-DP guarantee for DP Lloyd's algorithm with coreset-based sampling as follows. We apply Theorem 10 to the PDP profile derived in Proposition 13 and the coreset-based sampling distribution. For positive constants $a, b, s, t > 0$, this yields a $\psi$-PDP guarantee of the form

$$\psi(\mathbf{x}) = \log\left(1 + \left(\exp\left(\frac{a + t\|\mathbf{x}\|_2}{b + s\|\mathbf{x}\|_2^2}\right) - 1\right)\left(b + s\|\mathbf{x}\|_2^2\right)\right). \tag{5}$$

In order to derive an $\epsilon$-DP guarantee, we need to bound $\psi(\mathbf{x})$ over the domain of $\mathbf{x} \in \mathcal{B}_2(r)$. We observe that $\psi(\mathbf{x})$ depends on $\mathbf{x}$ only through $\|\mathbf{x}\|_2$. Therefore, we can bound $\psi(\mathbf{x})$ numerically by maximizing it over the domain $\|\mathbf{x}\|_2 \in [0, r]$ via grid search. The maximum always exists because $b$ and $s$ are strictly positive.

## 5 Experiments

We now evaluate our proposed sampling approaches (coreset-based and privacy-constrained) on the task of $k$-means clustering where we are interested in three objectives: privacy, efficiency, and accuracy. The point of the experiments is to investigate whether our proposed sampling strategies lead to improvements in terms of the three objectives when compared to uniform sampling. Our measure for efficiency is the subsample size $m$ produced by the sampling strategy. This is because Lloyd's algorithm scales linearly in $m$ (for fixed $k$ and $T$) and the computing time of the sampling itself is negligible in comparison. We measure accuracy via the $k$-means objective evaluated on the full data set and privacy as the $\epsilon$-DP guarantee of the sampled mechanism.

**Data.** We use the following eight real-world data sets: *Covertype* (Blackard & Dean, 1999) ($n = 581{,}012$, $d = 54$), *FMA*[2] (Defferrard et al., 2017) ($n = 106{,}574$, $d = 518$), *Ijcnn1*[3] (Chang & Lin, 2001) ($n = 49{,}990$, $d = 22$), *KDD-Protein*[4] ($n = 145{,}751$, $d = 74$), *MiniBooNE* (Dua & Graff, 2017) ($n = 130{,}064$, $d = 50$), *Pose*[5] (Catalin Ionescu, 2011; Ionescu et al., 2014) ($n = 35{,}832$, $d = 48$), *RNA* (Uzilov et al., 2006) ($n = 488{,}565$, $d = 8$), and *Song* (Bertin-Mahieux et al., 2011) ($n = 515{,}345$, $d = 90$).

We pre-process the data sets to ensure each data point has bounded $\ell_2$-norm. Following the common approach in differential privacy to bound contributions at a quantile (Abadi et al., 2016; Geyer et al., 2017; Amin et al., 2019), we set the $\ell_2$ cut-off point $r$ to the 97.5 percentile and discard the points whose norm exceeds $r$. Moreover, we center each data set since this is a prerequisite for the coreset-based sampling distribution, see Section 4.

**Setup.** The specific task we consider is $k$-means for which we use the weighted version of DP-Lloyd. We fix the number of iterations of DP-Lloyd to $T = 10$ and the number of clusters to $k = 25$. The scale parameters are set to $\beta_{\text{sum}} = \sqrt{\frac{Tr}{B}}\sqrt[3]{\frac{d}{2\rho}}$ and $\beta_{\text{count}} = \sqrt[3]{4d\rho^2}\beta_{\text{sum}}$, where $\rho = 0.225$ as suggested by Su et al. (2016). Here, $B$ is a constant controlling the noise scales that we select to achieve a specific target epsilon $\epsilon^\star \in \{0.5, 1, 3, 10, 30, 100, 300, 1000.0\}$ for a given (expected) subsample size $m$ and vice versa.

We evaluate the following three different importance samplers and use various sample sizes, *i.e.*, $m \in [3000, 75000]$, depending on the data set. For the coreset-based (**core**) sampling, the sampling distribution is $q_{\text{core}}(\mathbf{x}) = \lambda\frac{m}{n} + (1-\lambda)\frac{m\|\mathbf{x}\|_2^2}{n\tilde{\mathbf{x}}}$, where we set $\lambda = \frac{1}{2}$. The uniform (**unif**) sampling uses $q_{\text{unif}}(\mathbf{x}) = m/n$. Note

---

[2]`https://github.com/mdeff/fma`
[3]`https://www.csie.ntu.edu.tw/~cjlin/libsvmtools/datasets/`
[4]`http://osmot.cs.cornell.edu/kddcup/datasets.html`
[5]`http://vision.imar.ro/human3.6m/challenge_open.php`

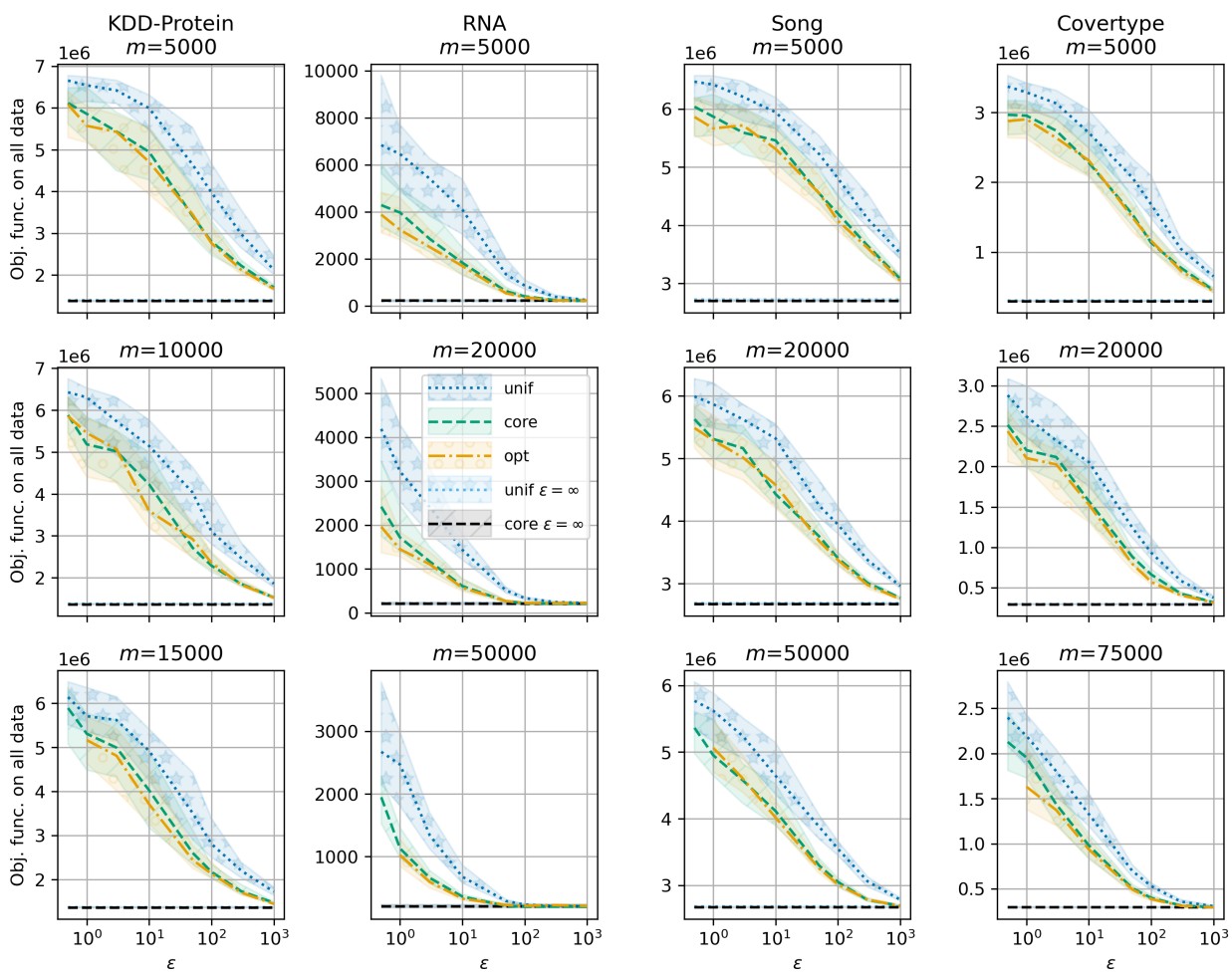

Figure 3: The trade-off between the privacy parameter $\epsilon$ and the total cost of DP-Lloyd on KDD-Protein, RNA, Song, and Covertype data. Non-private counterparts ($\epsilon = \infty$) are shown for reference. Lower is better on both axes. The $\epsilon$-axis is in log-scale.

that it is the same $q$ as in **core** but with $\lambda = 1$. For the privacy-constrained (**opt**) sampling, we compute $q_{\mathrm{priv}}(\mathbf{x})$ numerically via Algorithm 1 for the target $\epsilon^\star = (r/\beta_{\mathrm{sum}} + 1/\beta_{\mathrm{count}})T$ using the strong convexity constant from Equation (4).

Due to the stochasticity in the subsampling process and the noises within DP-Lloyd, we repeat each experiment 50 times with different seeds and report on median performances. In addition, we depict the 75% and 25% quartiles. Note that the weighted version of DP-Lloyd is only used when learning the cluster centers, *i.e.*, not for evaluation. Furthermore, we initialize the cluster centers as $k$ random data points. For all sampling-based approaches, we re-compute the objective function (cost) on all data after running DP-Lloyd on the sampled subset and report on the total cost scaled by the number of data points $n^{-1}$.

The code[6] is implemented in Python using numpy (Harris et al., 2020). Algorithm 1 is implemented in Rust. All experiments run on a dual AMD Epyc machine with $2 \times 64$ cores with 2.25 GHz and 2 TiB of memory.

**Results.** We first evaluate the trade-off between the privacy parameter $\epsilon$ and the total cost of DP-Lloyd. Figure 3 depicts the results for KDD-Protein, RNA, Song, and Covertype. Within the figure, each column

---

[6]https://github.com/smair/personalized-privacy-amplification-via-importance-sampling.

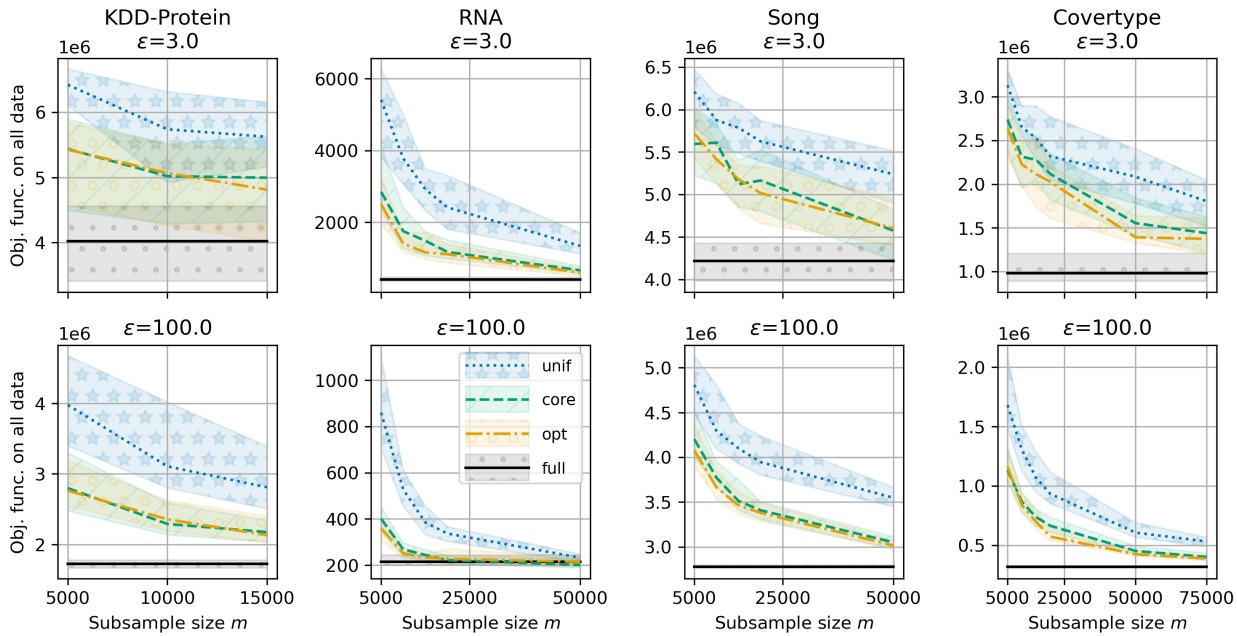

Figure 4: The trade-off between subsample size $m$ and the total cost of DP-Lloyd for fixed privacy parameters $\epsilon \in \{3, 100\}$ on KDD-Protein, RNA, Song, and Covertype data. The performance of DP-Lloyd on the full data is shown for reference. Lower is better on both axes.

corresponds to a data set and the rows show different subsampling sizes $m$. The curves are obtained by subsampling $m$ data points and training DP-Lloyd on the obtained subset for which we vary $B$ and re-compute the total cost, *i.e.*, the cost evaluated on all data. Thus, we see the performance DP-Lloyd as a function of the privacy parameter $\epsilon$. Note that the $x$-axis is in log-scale, shared column-wise, and that the optimal location is the bottom left since it yields a low cost and a low privacy parameter. Additional results on Ijcnn1, Pose, MiniBooNE, and FMA are shown in Figure 6 in Appendix B. The performance of a uniform subsample of the data set of size $m$ (**unif**, blue line) always yields the worst results. Our first proposed subsampling strategy **core** (green line) consistently outperforms **unif** as it yields a better cost-$\epsilon$-parameter trade-off. The other proposed sampler **opt** (yellow line) usually performs on-par or slightly better than **core**. Note that the privacy-constrained sampling strategy **opt** only optimizes two out of three objectives: privacy and efficiency but not accuracy. Although privacy and accuracy are often well aligned in this problem, there is no formal guarantee on the accuracy. In particular, the degree of alignment between privacy and accuracy depends on the specific mechanism and data set used. This is why its cost is not necessarily smaller than for **core**. Additionally, we show the non-private versions of **unif** and **core** (mostly overlapping) that correspond to the choice of $\epsilon = \infty$ as a reference. Note that the performance converges to the non-private version as we increase $\epsilon$. Note also that, as $\epsilon \to 0$, the cost approaches but stays below that of performing no learning at all. This can be seen by comparison to the $\tilde{x}$ column of Table 1: when the cluster centers are initialized to zero, the cost is precisely $\tilde{x}$, i.e., the average squared $\ell_2$ norm. The main take-away is that both our proposed sampling strategies consistently yield a better trade-off between the privacy parameter $\epsilon$ and the total cost of (DP-)$k$-means than **unif**.

Next, we evaluate the performance of the subsampling strategies as functions of the subsample size $m$, *i.e.*, the trade-off between sample size $m$ and the total cost of DP-Lloyd. For this scenario, we fix the privacy budget to either $\epsilon = 3$ or $\epsilon = 100$. Figure 4 depicts the results for the data sets KDD-Protein, RNA, Song, and Covertype. As expected, the total cost decreases as we increase the subsample size $m$. Moreover, we can see that **unif** (blue line) – once again – performs consistently worst. In contrast, the coreset-inspired sampling **core** (green line) and sampling using privacy-constrained weights **opt** (yellow line) yields lower total cost for the same $m$ and $\epsilon$. For reference, we also include the performance of DP-Lloyd using the same

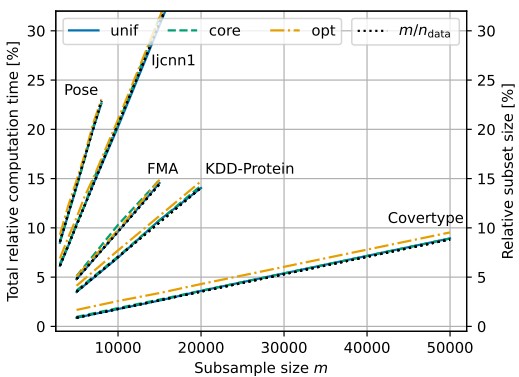 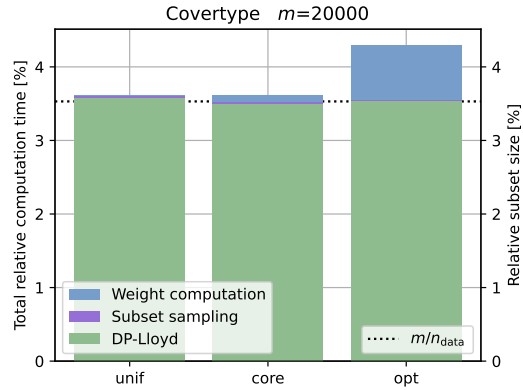

Figure 5: **Left**: Total relative computation times (left $y$-axis) and relative subset sizes (right $y$-axis) as functions of subsample sizes $m$ for five data sets. **Right**: Relative computation times decomposed in weight computation, sampling, and DP-Lloyd for the Covertype data and a subsample size of $m$=20,000.

privacy parameter $\epsilon$ but on all data, *i.e.*, without any subsampling, as a black line (**full**). Unsurprisingly, the total cost of the subsampling methods approaches the total cost of **full** as $m$ approaches $n$. Note that this happens faster for our subsampling methods than for **unif**.

Lastly, we measure the computation times of the different sampling strategies to confirm that they are short relative to the computation time of DP-Lloyd. This supplements the iteration complexity analysis from Proposition 12. Figure 5 (left) shows the total relative computation times (left $y$-axis), *i.e.*, the time to (i) compute the weights, (ii) subsample the data set, and (iii) compute DP-Lloyd for various subset sizes $m$, relative to computing DP-Lloyd on all data using $\epsilon = 3.0$ for five data sets. Each line in the figure refers to a combination of data set and sampling strategy. In addition, we show the fraction $m/n_{\mathrm{data}}$ (right $y$-axis) per data set as a black dotted line. To improve readability, we omit MiniBooNE, Song and RNA from the plot because they overlap with the lines of other data sets. For completeness, we show the numbers in tabular form in Appendix C for all data sets. A sampling strategy can be considered efficient if its relative vertical offset from the black dotted line is small. This is the case for all data sets and sampling strategies. Additionally, Figure 5 (right) depicts the decomposed computation times for the Covertype data set and a subsample size of $m$=20,000. We can see that the time needed to subsample the data set (violet) is negligible and that the time DP-Lloyd takes (green) is almost static. For this data set, the weight computation (blue) for the **opt** weight computation takes significantly more time than for **unif** and **core**. However, the difference is small relative to the runtime of DP-Lloyd on all data. Specifically, a subset size of $m$=20,000 amounts to approximately 3.5% of the Covertype data set. Using the **opt** subsampling strategy takes slightly more than 4% of the time DP-Lloyd takes on the full data set.

## 6 Related Work

The notion of personalized differential privacy is introduced by Jorgensen et al. (2015) and Ebadi et al. (2015), as well as by Alaggan et al. (2016) under the name of heterogeneous differential privacy. In Jorgensen et al. (2015), the privacy parameter is associated with a *user*, while it is associated with the *value* of a data point in the work of Ebadi et al. (2015) and ours. Jorgensen et al. (2015) achieve personalized DP by subsampling the data with heterogeneous probabilities, but without accounting for the bias introduced by the heterogeneity. Moreover, this privacy analysis is loose as it does not exploit the inherent heterogeneity of the original mechanism's privacy guarantee.

Recently, there has been renewed interest in PDP due to its connection to individual privacy accounting and fully adaptive composition (Feldman & Zrnic, 2021; Koskela et al., 2023; Yu et al., 2023). In this context, *privacy filters* have been proposed as a means to answer more queries about a data set by reducing a

PDP guarantee to its worst-case counterpart. Analogously, our privacy-constrained importance sampling distribution can be used to subsample a data set by reducing a PDP guarantee to its worst-case counterpart.

The idea of using importance sampling for differential privacy is not entirely new. Wei et al. (2022) propose a differentially private importance sampler for the mini-batch selection in differentially private stochastic gradient descent. Their sampling distribution resembles the *variance-optimal* distribution we display in Figure 2. The major drawback of this distribution is its intractability—it requires us to know the quantity we want to compute in the first place. Note that our privacy-constrained distribution is very close to the variance-optimal distribution while being efficient to compute. Moreover, their privacy analysis is restricted to the Gaussian mechanism and does not generalize to general DP or Rényi-DP mechanisms, because it is based on Mironov et al. (2019)'s analysis of the subsampled Gaussian mechanism.

The first differentially private version of $k$-means is introduced by Blum et al. (2005). They propose the Sub-Linear Queries (SuLQ) framework, which, for $k$-means, uses a noisy (Gaussian) estimate of the number of points per cluster and a noisy sum of distances. This is also reported by Su et al. (2016; 2017) which extend the analysis. Other variants of differentially private $k$-means were proposed, *e.g.*, by Balcan et al. (2017); Huang & Liu (2018); Stemmer & Kaplan (2018); Shechner et al. (2020); Ghazi et al. (2020); Nguyen et al. (2021); Cohen-Addad et al. (2022).

## 7 Conclusion

We introduced and analyzed Poisson importance sampling for subsampling differentially private mechanisms. We observed that, for typical mechanisms, privacy is well aligned with utility and at odds with sample size. Based on this insight, we proposed two importance sampling distributions: one that navigates the trade-off between privacy and sample size and another based on coresets which have strong utility guarantees. The empirical results suggest that both strategies have stronger privacy and utility than uniform sampling at any given sample size.

Promising directions for future work include extensions to $(\epsilon, \delta)$-DP or Rényi-DP as well as establishing formal utility guarantees via coresets. For the latter, recent work on confidence intervals for stratified sampling (Lin et al., 2024) might serve as a starting point. Moreover, Poisson importance sampling is directly applicable to a streaming setting, because it considers each data point separately. This provides an opportunity to improve efficiency further. In federated learning, Poisson importance sampling might be used to improve client selection (Zhang et al., 2024). Additionally, its connection to fairness could be explored where importance sampling can be used to mitigate bias (Wang et al., 2023).

## Acknowledgements

This work was partially supported by the Wallenberg AI, Autonomous Systems and Software Program (WASP) funded by the Knut and Alice Wallenberg Foundation. This research has been carried out as part of the Vinnova Competence Center for Trustworthy Edge Computing Systems and Applications at KTH Royal Institute of Technology.

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

## Appendix

- Section A discusses sufficient conditions under which importance sampling does or does not improve privacy.

- Section B includes experimental results on additional data sets.

- Section C contains details on the computation time measurements.

- Section D contains the proofs for Theorem 6, Proposition 12 and Proposition 13. Moreover, it contains Algorithm 2 which describes the weighted version of DP Lloyd.

## A    Can sampling improve privacy?

We have pointed out in Section 3 that, in importance sampling, the privacy loss is not necessarily reduced by decreasing the sampling probability. In this section, we make this statement more precise and provide formal conditions on the PDP profile under which weighted sampling does or does not improve privacy when reweighting is accounted for.

First, note that any mechanism $\mathcal{M}$ that simply ignores the weights $\{w_i\}_{i=1}^n$ has a weighted PDP profile $\epsilon(w, \mathbf{x})$ that is constant in $w$. In this case, Theorem 6 reduces to the established amplification by subsampling result (Proposition 3), which indeed implies that $\widehat{\mathcal{M}}$ satisfies a stronger privacy guarantee than $\mathcal{M}$. This is possible because such a mechanism is not invariant under splitting a weighted point $(w, \mathbf{x})$ into $w$ unweighted points $\{(1, \mathbf{x})\}_{i=1}^w$. The following proposition provides a more general sufficient condition under which $\widehat{\mathcal{M}}$ satisfies a stronger privacy guarantee than $\mathcal{M}$.

**Proposition 16.** *Let $\mathcal{M} : \mathcal{X}^* \times [1, \infty) \to \mathcal{Y}$ be an $\epsilon$-PDP mechanism, $\widetilde{\mathcal{M}}$ be its unweighted counterpart defined as $\widetilde{\mathcal{M}}(\{\mathbf{x}_i\}_{i=1}^n) = \mathcal{M}(\{(1, \mathbf{x}_i)\}_{i=1}^n)$, and $\tilde{\epsilon}(\mathbf{x}) = \epsilon(1, \mathbf{x})$ be the PDP profile of $\widetilde{\mathcal{M}}$. Furthermore, let $\epsilon$ be differentiable in $w$ and $\epsilon'(w, \mathbf{x}) = \partial \epsilon(w, \mathbf{x})/\partial w$. Then, there is a Poisson importance sampler $S(\cdot)$ for which the following holds. Let $\psi$ be the PDP profile of $\mathcal{M} \circ S$ implied by Theorem 6. For any $\mathbf{x} \in \mathcal{X}$, if $\epsilon'(1, \mathbf{x}) < 1 - e^{-\epsilon(1, \mathbf{x})}$, then $\psi(\mathbf{x}) < \tilde{\epsilon}(\mathbf{x})$.*

*Proof.* Let $\mathbf{x} \in \mathcal{X}$ be any data point for which $\epsilon'(1, \mathbf{x}) < 1 - e^{-\epsilon(1, \mathbf{x})}$ holds. We treat $\psi(\mathbf{x})$ as a function of the selection probability $q(\mathbf{x})$ and show that it is increasing at $q(\mathbf{x}) = 1$. We write $\psi_{q(\mathbf{x})}(\mathbf{x})$ to make the dependence on $q(\mathbf{x})$ explicit and define $\omega(w) = \exp(\psi_{1/w}(\mathbf{x})) - 1$. We have

$$\frac{\mathrm{d}\omega(1)}{\mathrm{d}w} = e^{\epsilon(1, \mathbf{x})}(\epsilon'(1, \mathbf{x}) - 1) + 1$$
$$< e^{\epsilon(1, \mathbf{x})}\left(\left(1 - e^{-\epsilon(1, \mathbf{x})}\right) - 1\right) + 1$$
$$= 0,$$

and, hence, $\mathrm{d}\psi_{q(\mathbf{x})}(\mathbf{x})/\mathrm{d}q(\mathbf{x}) > 0$ at $q(\mathbf{x}) = 1$. As a result, there is a probability $q(\mathbf{x}) < 1$ such that $\psi_{q(\mathbf{x})}(\mathbf{x}) < \psi_1(\mathbf{x}) = \tilde{\epsilon}(\mathbf{x})$. □

Consequently, if the condition $\epsilon'(1, \mathbf{x}) < 1 - e^{-\epsilon(1, \mathbf{x})}$ holds in a neighborhood around the maximizer $\mathbf{x}^\star = \arg\max_{\mathbf{x} \in \mathcal{X}} \tilde{\epsilon}(\mathbf{x})$, then $\widehat{\mathcal{M}}$ satisfies DP with a strictly smaller privacy parameter than $\mathcal{M}$.

However, for the following important class of mechanisms, $\widehat{\mathcal{M}}$ cannot satisfy a stronger privacy guarantee than $\mathcal{M}$.

**Proposition 17.** *Let $\mathcal{M}, \widetilde{\mathcal{M}}, \epsilon, \epsilon'$, and $\tilde{\epsilon}$ be as in Proposition 16, $S(\cdot)$ be any Poisson importance sampler, $\psi$ be the PDP profile of $\mathcal{M} \circ S$ implied by Theorem 6, and $\mathbf{x} \in \mathcal{X}$. If $\epsilon(w, \mathbf{x}) \leq w\epsilon'(w, \mathbf{x})$ for all $w \geq 1$, then $\psi(\mathbf{x}) \geq \tilde{\epsilon}(\mathbf{x})$.*

*Proof.* As in Proposition 16, the core idea is to treat $\psi(\mathbf{x})$ as a function of the selection probability $q(\mathbf{x})$. We show that $\psi(\mathbf{x})$ is non-increasing in $q(\mathbf{x})$ if the condition $\epsilon(w, \mathbf{x}) \leq w\epsilon'(w, \mathbf{x})$ is satisfied for all $w$.

Let $q(\mathbf{x})$ be the selection probability of $\mathbf{x}$ under $S$. We write $\psi_q(\mathbf{x})$ to make the dependence on $q(\mathbf{x})$ explicit. Let $\mathbf{x} \in \mathcal{X}$ be arbitrary but fixed and define $\omega(w) = \exp(\psi_{1/w}(\mathbf{x})) - 1$. We have

$$
\begin{aligned}
\frac{\mathrm{d}\omega(w)}{\mathrm{d}w} &= \frac{e^{\epsilon(w,\mathbf{x})}(w\epsilon'(w,\mathbf{x}) - 1) + 1}{w^2} \\
&\geq \frac{e^{\epsilon(w,\mathbf{x})}(\epsilon(w,\mathbf{x}) - 1) + 1}{w^2} \\
&\geq \frac{(\epsilon(w,\mathbf{x}) + 1)(\epsilon(w,\mathbf{x}) - 1) + 1}{w^2} \\
&\geq \frac{\epsilon(w,\mathbf{x})^2}{w^2} \\
&\geq 0,
\end{aligned}
$$

where we used the assumption $\epsilon(w,\mathbf{x}) \leq w\epsilon'(w,\mathbf{x})$ in the first inequality and the fact that $e^z \geq z + 1$ for all $z \geq 0$ in the second inequality. As a result, $\omega$ is minimized at $w = 1$ and, hence, $\psi_q$ is minimized at $q(\mathbf{x}) = 1$. To complete the proof, observe that $\psi_q(\mathbf{x}) = \tilde{\epsilon}(\mathbf{x})$ for $q(\mathbf{x}) = 1$. $\qquad\square$

For instance, the condition $\epsilon(w,\mathbf{x}) \leq w\epsilon'(w,\mathbf{x})$ is satisfied everywhere by any profile of the form $\epsilon(w,\mathbf{x}) = f(\mathbf{x})w^p$, where $p \geq 1$ and $f$ is any non-negative function that does not depend on $w$. This includes any mechanism that is invariant under splitting a weighted point $(w,\mathbf{x})$ into $w$ unweighted points $\{(1,\mathbf{x})\}_{i=1}^{w}$ since group privacy implies a linear PDP profile in this case. All mechanisms considered in this paper satisfy this invariance.

It is important to note that, even in a case where we cannot hope to improve upon the original mechanism, it is still possible to obtain a stronger privacy amplification than with uniform subsampling at the same sampling rate. Indeed, the uniform distribution is never optimal unless the PDP profile of the original mechanism is constant.

# B    Results on the Remaining Data Sets

Table 1: Information on the data sets. Remember that $n, d, r$ and $\tilde{x}$ denote the sample size, dimensionality, largest $\ell_2$-norm, and average squared $\ell_2$-norm, respectively.

| | Data Set | $n$ (after outlier removal) | $d$ | $r$ | $\tilde{x}$ |
|---|---|---|---|---|---|
| Main Paper | KDD-Protein | 142,107 | 74 | 8020.65 | 7031359.55 |
| | RNA | 476,350 | 8 | 420.67 | 24902.47 |
| | Song | 502,461 | 90 | 7507.16 | 6769375.49 |
| | Covertype | 566,486 | 54 | 4489.17 | 3839798.28 |
| Appendix | Ijcnn1 | 48,740 | 22 | 1.51 | 1.26 |
| | Pose | 34,936 | 48 | 2250.64 | 1249739.50 |
| | MiniBooNE | 126,812 | 50 | 2213.05 | 295947.13 |
| | FMA | 103,909 | 518 | 6020.23 | 5838871.88 |

Figure 6 depicts the results for the remaining data sets Ijcnn1, Pose, MiniBooNE, and FMA. Note that those data sets are smaller in terms of the number of data points as the data sets shown in Figure 3, see Table 1. As seen in Figure 6, we can observe the smallest difference among the subsampling strategies across all data sets occurs for Ijcnn1, especially for the $m=15,000$ case. All subsampling strategies behave alike, except for the $m=10,000$ case where **unif** is worse than **core** and **opt**. On Pose, MiniBooNE, and FMA, our subsampling strategies consistently outperform **unif**. In addition, Figure 7 depicts the cost-sample-size-$m$ ratios for the remaining data sets Ijcnn1, Pose, MiniBooNE, and FMA for fixed privacy budgets of $\epsilon = 3$ and $\epsilon = 100$. Once again, the only data set in which no clear improvement is visible is Ijcnn1.

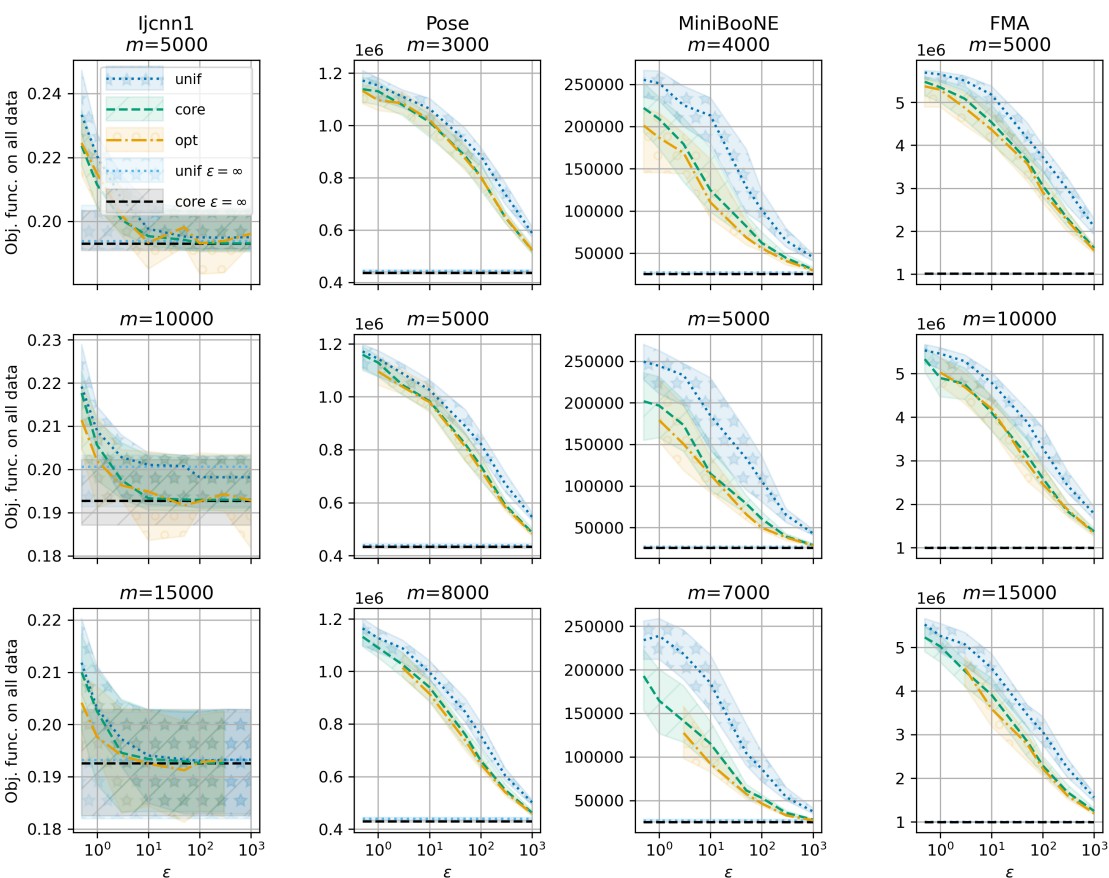

Figure 6: The trade-off between the privacy parameter $\epsilon$ and the total cost of DP-Lloyd on Ijcnn1, Pose, MiniBooNE, and FMA data. Non-private counterparts ($\epsilon = \infty$) are shown for reference. Lower is better on both axes. The $\epsilon$-axis is in log-scale.

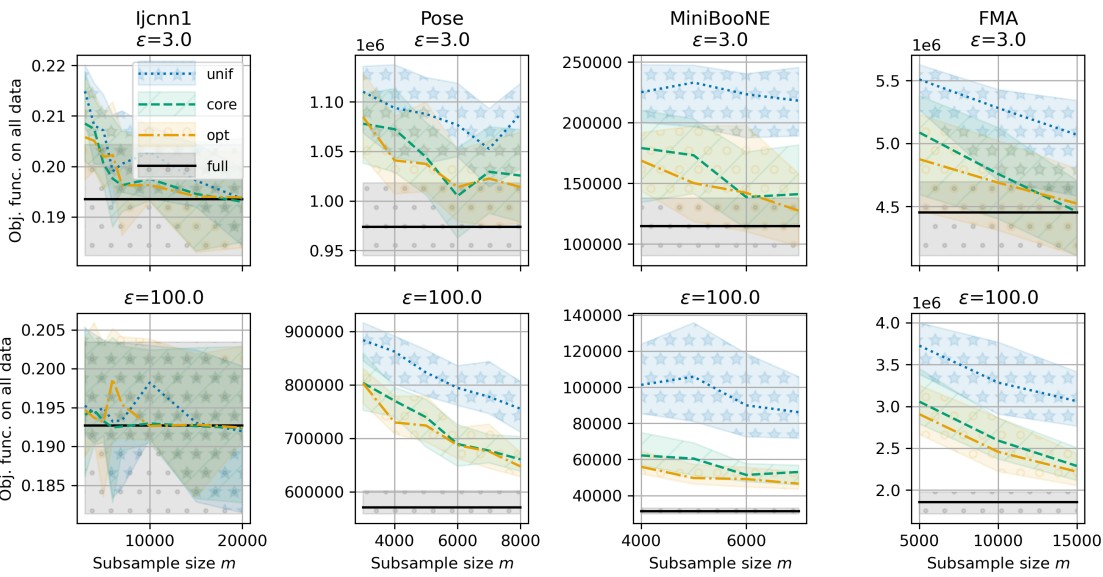

Figure 7: The trade-off between sample size $m$ and the total cost of DP-Lloyd for fixed privacy parameters $\epsilon \in \{3, 100\}$ on Ijcnn1, Pose, MiniBooNE, and FMA. The performance of DP-Lloyd on the full data is shown for reference. Lower is better on both axes.

## C  Detailed Timing Results

Table 2: The exact numbers of Figure 5 for the first four data sets. Times are in seconds.

| Data Set | Sampling | $m$ | $t_{\text{weights}}$ | $t_{\text{sampling}}$ | $t_{\text{DP-Lloyd}}$ | $t_{\text{total}}$ | rel. tot. time [%] | $\frac{m}{n}$ [%] |
|---|---|---|---|---|---|---|---|---|
| KDD-Protein | full | $n$ | - | - | 15.69 | 15.69 | 100 | 100 |
| KDD-Protein | unif | 5000 | 0.0005 | 0.0026 | 0.54 | 0.55 | 3.49 | 3.52 |
| KDD-Protein | unif | 10,000 | 0.0004 | 0.0029 | 1.09 | 1.10 | 6.99 | 7.04 |
| KDD-Protein | unif | 15,000 | 0.0004 | 0.0035 | 1.69 | 1.69 | 10.79 | 10.56 |
| KDD-Protein | unif | 20,000 | 0.0004 | 0.0042 | 2.20 | 2.20 | 14.04 | 14.07 |
| KDD-Protein | core | 5000 | 0.0198 | 0.0025 | 0.55 | 0.57 | 3.64 | 3.52 |
| KDD-Protein | core | 10,000 | 0.0187 | 0.0030 | 1.09 | 1.11 | 7.10 | 7.04 |
| KDD-Protein | core | 15,000 | 0.0182 | 0.0034 | 1.66 | 1.69 | 10.75 | 10.56 |
| KDD-Protein | core | 20,000 | 0.0185 | 0.0040 | 2.22 | 2.24 | 14.29 | 14.07 |
| KDD-Protein | opt | 5000 | 0.1112 | 0.0023 | 0.53 | 0.65 | 4.12 | 3.52 |
| KDD-Protein | opt | 10,000 | 0.1076 | 0.0031 | 1.10 | 1.21 | 7.72 | 7.04 |
| KDD-Protein | opt | 15,000 | 0.1134 | 0.0034 | 1.64 | 1.76 | 11.22 | 10.56 |
| KDD-Protein | opt | 20,000 | 0.1051 | 0.0039 | 2.20 | 2.31 | 14.73 | 14.07 |
| RNA | full | $n$ | - | - | 42.24 | 42.24 | 100 | 100 |
| RNA | unif | 5000 | 0.0004 | 0.0051 | 0.43 | 0.44 | 1.04 | 1.05 |
| RNA | unif | 10,000 | 0.0006 | 0.0056 | 0.88 | 0.88 | 2.09 | 2.10 |
| RNA | unif | 15,000 | 0.0005 | 0.0060 | 1.32 | 1.32 | 3.14 | 3.15 |
| RNA | unif | 20,000 | 0.0006 | 0.0064 | 1.73 | 1.74 | 4.12 | 4.20 |
| RNA | unif | 50,000 | 0.0005 | 0.0082 | 4.35 | 4.36 | 10.33 | 10.50 |
| RNA | core | 5000 | 0.0153 | 0.0051 | 0.44 | 0.46 | 1.09 | 1.05 |
| RNA | core | 10,000 | 0.0149 | 0.0057 | 0.87 | 0.89 | 2.12 | 2.10 |
| RNA | core | 15,000 | 0.0136 | 0.0061 | 1.33 | 1.35 | 3.19 | 3.15 |
| RNA | core | 20,000 | 0.0153 | 0.0066 | 1.74 | 1.76 | 4.18 | 4.20 |
| RNA | core | 50,000 | 0.0160 | 0.0080 | 4.34 | 4.37 | 10.34 | 10.50 |
| RNA | opt | 5000 | 0.3956 | 0.0051 | 0.44 | 0.84 | 1.98 | 1.05 |
| RNA | opt | 10,000 | 0.3865 | 0.0055 | 0.88 | 1.27 | 3.02 | 2.10 |
| RNA | opt | 15,000 | 0.3817 | 0.0061 | 1.30 | 1.69 | 3.99 | 3.15 |
| RNA | opt | 20,000 | 0.3776 | 0.0066 | 1.73 | 2.12 | 5.01 | 4.20 |
| RNA | opt | 50,000 | 0.3576 | 0.0085 | 4.40 | 4.77 | 11.28 | 10.50 |
| Song | full | $n$ | - | - | 59.47 | 59.47 | 100 | 100 |
| Song | unif | 5000 | 0.0016 | 0.0074 | 0.56 | 0.57 | 0.96 | 1.00 |
| Song | unif | 10,000 | 0.0018 | 0.0077 | 1.14 | 1.15 | 1.94 | 1.99 |
| Song | unif | 15,000 | 0.0017 | 0.0077 | 1.76 | 1.77 | 2.97 | 2.99 |
| Song | unif | 20,000 | 0.0015 | 0.0084 | 2.29 | 2.30 | 3.86 | 3.98 |
| Song | unif | 50,000 | 0.0017 | 0.0146 | 5.75 | 5.77 | 9.70 | 9.95 |
| Song | core | 5000 | 0.0748 | 0.0067 | 0.56 | 0.64 | 1.08 | 1.00 |
| Song | core | 10,000 | 0.0774 | 0.0076 | 1.15 | 1.24 | 2.08 | 1.99 |
| Song | core | 15,000 | 0.0765 | 0.0084 | 1.74 | 1.82 | 3.06 | 2.99 |
| Song | core | 20,000 | 0.0750 | 0.0091 | 2.31 | 2.39 | 4.03 | 3.98 |
| Song | core | 50,000 | 0.0780 | 0.0145 | 5.80 | 5.89 | 9.90 | 9.95 |
| Song | opt | 5000 | 0.4353 | 0.0063 | 0.57 | 1.01 | 1.69 | 1.00 |
| Song | opt | 10,000 | 0.4020 | 0.0075 | 1.16 | 1.57 | 2.64 | 1.99 |
| Song | opt | 15,000 | 0.4073 | 0.0083 | 1.73 | 2.15 | 3.61 | 2.99 |
| Song | opt | 20,000 | 0.3945 | 0.0088 | 2.31 | 2.71 | 4.56 | 3.98 |
| Song | opt | 50,000 | 0.3789 | 0.0146 | 5.85 | 6.25 | 10.51 | 9.95 |
| Covertype | full | $n$ | - | - | 59.91 | 59.91 | 100 | 100 |
| Covertype | unif | 5000 | 0.0013 | 0.0062 | 0.52 | 0.53 | 0.88 | 0.88 |
| Covertype | unif | 10,000 | 0.0019 | 0.0070 | 1.05 | 1.06 | 1.76 | 1.77 |
| Covertype | unif | 15,000 | 0.0013 | 0.0077 | 1.56 | 1.57 | 2.63 | 2.65 |
| Covertype | unif | 20,000 | 0.0014 | 0.0082 | 2.15 | 2.16 | 3.61 | 3.53 |
| Covertype | unif | 50,000 | 0.0014 | 0.0115 | 5.34 | 5.35 | 8.93 | 8.83 |
| Covertype | core | 5000 | 0.0558 | 0.0062 | 0.52 | 0.58 | 0.97 | 0.88 |
| Covertype | core | 10,000 | 0.0567 | 0.0068 | 1.05 | 1.11 | 1.86 | 1.77 |
| Covertype | core | 15,000 | 0.0563 | 0.0076 | 1.57 | 1.63 | 2.72 | 2.65 |
| Covertype | core | 20,000 | 0.0569 | 0.0082 | 2.10 | 2.17 | 3.62 | 3.53 |
| Covertype | core | 50,000 | 0.0567 | 0.0115 | 5.28 | 5.34 | 8.92 | 8.83 |
| Covertype | opt | 5000 | 0.4684 | 0.0063 | 0.53 | 1.00 | 1.67 | 0.88 |
| Covertype | opt | 10,000 | 0.4614 | 0.0069 | 1.07 | 1.54 | 2.57 | 1.77 |
| Covertype | opt | 15,000 | 0.4440 | 0.0077 | 1.58 | 2.03 | 3.38 | 2.65 |
| Covertype | opt | 20,000 | 0.4404 | 0.0082 | 2.13 | 2.57 | 4.30 | 3.53 |
| Covertype | opt | 50,000 | 0.4237 | 0.0115 | 5.28 | 5.71 | 9.54 | 8.83 |

Table 3: The exact numbers of Figure 5 for the last four data sets. Times are in seconds.

| Data Set | Sampling | $m$ | $t_{\text{weights}}$ | $t_{\text{sampling}}$ | $t_{\text{DP-Lloyd}}$ | $t_{\text{total}}$ | rel. tot. time [%] | $\frac{m}{n}$ [%] |
|---|---|---|---|---|---|---|---|---|
| Ijcnn1 | full | $n$ | - | - | 4.68 | 4.68 | 100 | 100 |
| Ijcnn1 | unif | 3000 | 0.0001 | 0.0008 | 0.29 | 0.29 | 6.15 | 6.16 |
| Ijcnn1 | unif | 4000 | 0.0001 | 0.0008 | 0.38 | 0.38 | 8.15 | 8.21 |
| Ijcnn1 | unif | 5000 | 0.0001 | 0.0009 | 0.48 | 0.49 | 10.37 | 10.26 |
| Ijcnn1 | unif | 6000 | 0.0001 | 0.0010 | 0.58 | 0.58 | 12.33 | 12.31 |
| Ijcnn1 | unif | 7000 | 0.0001 | 0.0011 | 0.67 | 0.67 | 14.38 | 14.36 |
| Ijcnn1 | unif | 10,000 | 0.0001 | 0.0012 | 0.95 | 0.95 | 20.26 | 20.52 |
| Ijcnn1 | unif | 15,000 | 0.0001 | 0.0015 | 1.43 | 1.43 | 30.55 | 30.78 |
| Ijcnn1 | unif | 20,000 | 0.0002 | 0.0017 | 1.92 | 1.92 | 41.13 | 41.03 |
| Ijcnn1 | core | 3000 | 0.0022 | 0.0008 | 0.30 | 0.30 | 6.37 | 6.16 |
| Ijcnn1 | core | 4000 | 0.0022 | 0.0009 | 0.38 | 0.39 | 8.27 | 8.21 |
| Ijcnn1 | core | 5000 | 0.0026 | 0.0009 | 0.48 | 0.48 | 10.33 | 10.26 |
| Ijcnn1 | core | 6000 | 0.0021 | 0.0010 | 0.57 | 0.58 | 12.31 | 12.31 |
| Ijcnn1 | core | 7000 | 0.0021 | 0.0013 | 0.69 | 0.70 | 14.86 | 14.36 |
| Ijcnn1 | core | 10,000 | 0.0022 | 0.0012 | 0.96 | 0.96 | 20.49 | 20.52 |
| Ijcnn1 | core | 15,000 | 0.0021 | 0.0015 | 1.46 | 1.47 | 31.36 | 30.78 |
| Ijcnn1 | core | 20,000 | 0.0022 | 0.0017 | 1.93 | 1.94 | 41.40 | 41.03 |
| Ijcnn1 | opt | 3000 | 0.0347 | 0.0008 | 0.29 | 0.32 | 6.90 | 6.16 |
| Ijcnn1 | opt | 4000 | 0.0341 | 0.0008 | 0.39 | 0.42 | 9.03 | 8.21 |
| Ijcnn1 | opt | 5000 | 0.0340 | 0.0009 | 0.48 | 0.52 | 11.02 | 10.26 |
| Ijcnn1 | opt | 6000 | 0.0338 | 0.0009 | 0.58 | 0.62 | 13.23 | 12.31 |
| Ijcnn1 | opt | 7000 | 0.0331 | 0.0010 | 0.67 | 0.70 | 15.01 | 14.36 |
| Ijcnn1 | opt | 10,000 | 0.0329 | 0.0012 | 0.95 | 0.98 | 20.99 | 20.52 |
| Ijcnn1 | opt | 15,000 | 0.0318 | 0.0015 | 1.43 | 1.47 | 31.36 | 30.78 |
| Ijcnn1 | opt | 20,000 | 0.0319 | 0.0017 | 1.91 | 1.94 | 41.51 | 41.03 |
| Pose | full | $n$ | - | - | 3.59 | 3.59 | 100 | 100 |
| Pose | unif | 3000 | 0.0001 | 0.0007 | 0.30 | 0.30 | 8.49 | 8.59 |
| Pose | unif | 4000 | 0.0001 | 0.0008 | 0.41 | 0.41 | 11.35 | 11.45 |
| Pose | unif | 5000 | 0.0001 | 0.0009 | 0.51 | 0.51 | 14.15 | 14.31 |
| Pose | unif | 6000 | 0.0001 | 0.0010 | 0.62 | 0.62 | 17.19 | 17.17 |
| Pose | unif | 7000 | 0.0001 | 0.0011 | 0.72 | 0.72 | 20.04 | 20.04 |
| Pose | unif | 8000 | 0.0001 | 0.0017 | 0.81 | 0.81 | 22.64 | 22.90 |
| Pose | core | 3000 | 0.0029 | 0.0007 | 0.31 | 0.31 | 8.65 | 8.59 |
| Pose | core | 4000 | 0.0026 | 0.0008 | 0.40 | 0.41 | 11.37 | 11.45 |
| Pose | core | 5000 | 0.0026 | 0.0009 | 0.50 | 0.51 | 14.15 | 14.31 |
| Pose | core | 6000 | 0.0025 | 0.0010 | 0.61 | 0.61 | 17.02 | 17.17 |
| Pose | core | 7000 | 0.0026 | 0.0011 | 0.72 | 0.73 | 20.24 | 20.04 |
| Pose | core | 8000 | 0.0028 | 0.0011 | 0.81 | 0.81 | 22.68 | 22.90 |
| Pose | opt | 3000 | 0.0257 | 0.0008 | 0.31 | 0.33 | 9.30 | 8.59 |
| Pose | opt | 4000 | 0.0255 | 0.0008 | 0.41 | 0.44 | 12.24 | 11.45 |
| Pose | opt | 5000 | 0.0253 | 0.0009 | 0.50 | 0.53 | 14.79 | 14.31 |
| Pose | opt | 6000 | 0.0252 | 0.0010 | 0.60 | 0.63 | 17.46 | 17.17 |
| Pose | opt | 7000 | 0.0247 | 0.0011 | 0.71 | 0.74 | 20.59 | 20.04 |
| Pose | opt | 8000 | 0.0245 | 0.0011 | 0.80 | 0.83 | 23.05 | 22.90 |
| MiniBooNE | full | $n$ | - | - | 13.16 | 13.16 | 100 | 100 |
| MiniBooNE | unif | 4000 | 0.0004 | 0.0019 | 0.41 | 0.41 | 3.12 | 3.15 |
| MiniBooNE | unif | 5000 | 0.0004 | 0.0021 | 0.51 | 0.51 | 3.91 | 3.94 |
| MiniBooNE | unif | 6000 | 0.0004 | 0.0022 | 0.61 | 0.61 | 4.66 | 4.73 |
| MiniBooNE | unif | 7000 | 0.0004 | 0.0025 | 0.72 | 0.72 | 5.48 | 5.52 |
| MiniBooNE | core | 4000 | 0.0122 | 0.0020 | 0.41 | 0.42 | 3.19 | 3.15 |
| MiniBooNE | core | 5000 | 0.0127 | 0.0022 | 0.52 | 0.54 | 4.08 | 3.94 |
| MiniBooNE | core | 6000 | 0.0128 | 0.0022 | 0.62 | 0.64 | 4.85 | 4.73 |
| MiniBooNE | core | 7000 | 0.0129 | 0.0030 | 0.73 | 0.75 | 5.69 | 5.52 |
| MiniBooNE | opt | 4000 | 0.1000 | 0.0019 | 0.41 | 0.51 | 3.88 | 3.15 |
| MiniBooNE | opt | 5000 | 0.1019 | 0.0021 | 0.52 | 0.62 | 4.73 | 3.94 |
| MiniBooNE | opt | 6000 | 0.1000 | 0.0023 | 0.61 | 0.71 | 5.39 | 4.73 |
| MiniBooNE | opt | 7000 | 0.0976 | 0.0025 | 0.73 | 0.83 | 6.33 | 5.52 |
| FMA | full | $n$ | - | - | 26.65 | 26.65 | 100 | 100 |
| FMA | unif | 5000 | 0.0002 | 0.0040 | 1.27 | 1.28 | 4.79 | 4.81 |
| FMA | unif | 10,000 | 0.0002 | 0.0068 | 2.56 | 2.57 | 9.63 | 9.62 |
| FMA | unif | 15,000 | 0.0002 | 0.0145 | 3.86 | 3.88 | 14.55 | 14.44 |
| FMA | core | 5000 | 0.0798 | 0.0040 | 1.28 | 1.36 | 5.10 | 4.81 |
| FMA | core | 10,000 | 0.0798 | 0.0112 | 2.66 | 2.75 | 10.31 | 9.62 |
| FMA | core | 15,000 | 0.0859 | 0.0133 | 3.86 | 3.96 | 14.87 | 14.44 |
| FMA | opt | 5000 | 0.0785 | 0.0041 | 1.27 | 1.35 | 5.06 | 4.81 |
| FMA | opt | 10,000 | 0.0760 | 0.0069 | 2.53 | 2.61 | 9.80 | 9.62 |
| FMA | opt | 15,000 | 0.0754 | 0.0124 | 3.88 | 3.97 | 14.89 | 14.44 |

# D   Proofs

This section contains the proofs of all our theoretical results. For completeness, we do not only restate the theorems and propositions, but also the used algorithms, assumptions, and problems.

## D.1   Theorem 6

**Theorem 6** (Amplification by Importance Sampling). *Let $\mathcal{M} : [1, \infty) \times \mathcal{X}^* \to \mathcal{Y}$ be an $\epsilon$-PDP mechanism that operates on weighted data sets, $q \colon \mathcal{X} \to [0, 1]$ be a function, and $S_q(\cdot)$ be a Poisson importance sampler for $q$. The mechanism $\widehat{\mathcal{M}} = \mathcal{M} \circ S_q$ satisfies $\psi$-PDP with*

$$\psi(\mathbf{x}) = \log\Big(1 + q(\mathbf{x})\big(e^{\epsilon(w, \mathbf{x})} - 1\big)\Big), \quad \text{where} \quad w = \frac{1}{q(\mathbf{x})}. \tag{1}$$

*Proof.* The proof is analogous to the proof of the regular privacy amplification theorem with $\delta = 0$, see, *e.g.*, Steinke (2022, Theorem 29). Let $\mathcal{A}$ be any measurable subset from the probability space of $\mathcal{M}(\mathcal{D})$. We begin by defining the functions $P(\mathcal{Z}) = \Pr\Big[\widehat{\mathcal{M}}(\mathcal{D}) \in \mathcal{A} \mid S_q(\mathcal{D}) = \mathcal{Z}\Big]$ and $P'(\mathcal{Z}) = \Pr\Big[\widehat{\mathcal{M}}(\mathcal{D}') \in \mathcal{A} \mid S_q(\mathcal{D}') = \mathcal{Z}\Big]$. Let $\mathcal{D} = \mathcal{D}' \cup \{\mathbf{x}\}$ for some $\mathbf{x} \in \mathcal{X}$. Note that

$$\begin{aligned}
\Pr\Big[\widehat{\mathcal{M}}(\mathcal{D}) \in \mathcal{A}\Big] &= \mathbb{E}[P(S_q(\mathcal{D}))] \\
&= q(\mathbf{x})\mathbb{E}[P(S_q(\mathcal{D})) \mid \mathbf{x} \in S_q(\mathcal{D})] + (1 - q(\mathbf{x}))\mathbb{E}[P(S_q(\mathcal{D})) \mid \mathbf{x} \notin S_q(\mathcal{D})].
\end{aligned} \tag{6}$$

We can analyze the events $\mathbf{x} \in S_q(\mathcal{D})$ and $\mathbf{x} \notin S_q(\mathcal{D})$ separately as follows. Conditioned on the event $\mathbf{x} \notin S_q(\mathcal{D})$, the distributions of $S_q(\mathcal{D})$ and $S_q(\mathcal{D}')$ are identical since all selection events $\gamma_i$ are independent:

$$\mathbb{E}[P(S_q(\mathcal{D})) \mid \mathbf{x} \notin S_q(\mathcal{D})] = \mathbb{E}[P(S_q(\mathcal{D}')) \mid \mathbf{x} \notin S_q(\mathcal{D})] = \mathbb{E}[P'(S_q(\mathcal{D}'))]. \tag{7}$$

On the other hand, conditioned on the event $\mathbf{x} \in S_q(\mathcal{D})$, the sets $S_q(\mathcal{D})$ and $S_q(\mathcal{D}) \setminus \{\mathbf{x}\}$ are neighboring, and the distributions of $S_q(\mathcal{D}) \setminus \{\mathbf{x}\}$ and $S_q(\mathcal{D}')$ are identical. We can use the PDP profile $\epsilon$ to bound

$$\begin{aligned}
\mathbb{E}[P(S_q(\mathcal{D})) \mid \mathbf{x} \in S_q(\mathcal{D})] &\leq \mathbb{E}\Big[e^{\epsilon(1/q(\mathbf{x}), \mathbf{x})} P'(S_q(\mathcal{D}')) \mid \mathbf{x} \in S_q(\mathcal{D})\Big] \\
&= e^{\epsilon(1/q(\mathbf{x}), \mathbf{x})}\mathbb{E}[P'(S_q(\mathcal{D}'))].
\end{aligned} \tag{8}$$

Plugging Equations (7) and (8) into Equation (6) yields

$$\begin{aligned}
\Pr\Big[\widehat{\mathcal{M}}(\mathcal{D}) \in \mathcal{A}\Big] &\leq (1 - q(\mathbf{x}))\mathbb{E}[P'(S_q(\mathcal{D}'))] + q(\mathbf{x})e^{\epsilon(1/q(\mathbf{x}), \mathbf{x})}\mathbb{E}[P'(S_q(\mathcal{D}'))] \\
&= \Big(1 + q(\mathbf{x})\big(e^{1/q(\mathbf{x})} - 1\big)\Big)\mathbb{E}[P'(S_q(\mathcal{D}'))] \\
&= \Big(1 + q(\mathbf{x})\big(e^{1/q(\mathbf{x})} - 1\big)\Big)\Pr\Big[\widehat{\mathcal{M}}(\mathcal{D}') \in \mathcal{A}\Big].
\end{aligned}$$

An identical argument shows that

$$\Pr\Big[\widehat{\mathcal{M}}(\mathcal{D}) \in \mathcal{A}\Big] \geq \frac{1}{1 + q(\mathbf{x})\big(e^{1/q(\mathbf{x})} - 1\big)} \Pr\Big[\widehat{\mathcal{M}}(\mathcal{D}') \in \mathcal{A}\Big].$$

$\square$

### D.2 Theorem 10

**Problem 7** (Privacy-constrained sampling). *For a PDP profile $\epsilon\colon [1,\infty)\times \mathcal{X}\to\mathbb{R}_{\geq 0}$, a target privacy guarantee $\epsilon^\star$, and a data set $\mathcal{D}\subseteq\mathcal{X}$ of size $n$, we define the optimization problem for privacy-constrained sampling as*

$$\underset{w_1,\ldots,w_n}{\text{minimize}} \quad \sum_{i=1}^{n} \frac{1}{w_i} \tag{2a}$$

$$\text{subject to} \quad \log\left(1 + \frac{1}{w_i}\left(e^{\epsilon(w_i,\mathbf{x}_i)} - 1\right)\right) \leq \epsilon^\star \qquad \text{for all } i, \tag{2b}$$

$$w_i \geq 1, \qquad\qquad\qquad\qquad \text{for all } i. \tag{2c}$$

**Assumption 8.** For all $\mathbf{x}\in\mathcal{X}$, $\epsilon^\star \geq \epsilon(1,\mathbf{x})$.

**Assumption 9.** For all $\mathbf{x}\in\mathcal{X}$, there is a constant $v_{\mathbf{x}}\geq 1$ such that $\epsilon(w,\mathbf{x}) > \log(1 + w(e^{\epsilon^\star}-1))$ for all $w \geq v_{\mathbf{x}}$.

**Assumption 11.** The function $\epsilon(w,\mathbf{x})$ is differentiable w.r.t. $w$ and $\exp\circ\,\epsilon$ is $\mu_{\mathbf{x}}$-strongly convex in $w$ for all $\mathbf{x}$.

---

**Algorithm 3.2** Optimization for privacy-constrained importance weights

---

1: **Input:** Data set $\mathcal{D} = \{\mathbf{x}_1,\ldots,\mathbf{x}_n\}$, target privacy guarantee $\epsilon^\star > 0$, PDP profile $\epsilon$, its derivative $\epsilon'$ with respect to $w$, and strong convexity constants $\mu_1,\ldots,\mu_n$
2: **Output:** Importance weights $w_1,\ldots,w_n$
3: **for** $i = 1,\ldots,n$ **do**
4:     Define $g_i(w) = \frac{1}{w}\left(e^{\epsilon(w,\mathbf{x}_i)}-1\right) - \left(e^{\epsilon^\star}-1\right)$
5:     $\bar{v}_i \leftarrow \min\{e^{\epsilon(1,\mathbf{x}_i)} + \mu_i/2,\; \epsilon'(1,\mathbf{x}_i)e^{\epsilon(1,\mathbf{x}_i)} + 1\}$
6:     $b_i \leftarrow 2(e^{\epsilon^\star} - \bar{v}_i)/\mu_i + 1$
7:     **if** $\epsilon(1,\mathbf{x}_i) = \epsilon^\star$ and $\epsilon'(1,\mathbf{x}_i) < 0$ **then**
8:         $w_i \leftarrow$ Bisect $g_i$ with initial bracket $(1, b_i]$
9:     **else**
10:         $w_i \leftarrow$ Bisect $g_i$ with initial bracket $[1, b_i]$
11:     **end if**
12: **end for**

---

**Theorem 10.** *Let Assumptions 8 and 9 be satisfied. There is a data set-independent function $w\colon \mathcal{X}\to[1,\infty)$, such that, for all data sets $\mathcal{D}\in\mathcal{X}^*$, Problem 7 has a unique solution $w^\star(\mathcal{D}) = (w_1^\star(\mathcal{D}),\ldots,w_n^\star(\mathcal{D}))$ of the form $w_i^\star(\mathcal{D}) = w(\mathbf{x}_i)$. Furthermore, let $\mathcal{M}$ be a mechanism that admits the PDP profile $\epsilon$ and $S_q$ be a Poisson importance sampler for $q(\mathbf{x}) = 1/w(\mathbf{x})$. Then, $\mathcal{M}\circ S_q$ satisfies $\epsilon^\star$-DP.*

*Proof.* Since the objective is additive over $w_i$ and each constraint only affects one $w_i$, we can consider each $w_i$ separately. An equivalent formulation of the problem is

$$\underset{w_i}{\text{maximize}} \quad w_i$$

$$\text{subject to} \quad \log\left(1 + \frac{1}{w_i}\left(e^{\epsilon(w_i,\mathbf{x}_i)} - 1\right)\right) \leq \epsilon^\star$$

$$w_i \geq 1.$$

Assumption 8 guarantees that the feasible region is non-empty and Assumption 9 guarantees that it is bounded. Since the objective is strictly monotonic, the solution must be unique. $\qquad\square$

### D.3 Proposition 12

**Proposition 12.** *Let Assumptions 8 and 11 be satisfied. Algorithm 1 solves Problem 7 up to accuracy $\alpha$ with at most $\sum_{\mathbf{x}\in\mathcal{D}} \log_2\lceil(e^{\epsilon^\star} - \bar{v}_{\mathbf{x}})/(\alpha\mu_{\mathbf{x}})\rceil$ evaluations of $\epsilon(w,\mathbf{x})$, where $\epsilon'(w,\mathbf{x}) = \partial/\partial w\,\epsilon(w,\mathbf{x})$ and $\bar{v}_{\mathbf{x}} = \min\{e^{\epsilon(1,\mathbf{x})} + \mu_{\mathbf{x}}/2,\; \epsilon'(1,\mathbf{x})e^{\epsilon(1,\mathbf{x})} + 1\}$.*

Before beginning the proof, we state the definition of strong convexity for completeness.

**Definition 18** (Strong convexity). Let $\mu > 0$. A differentiable function $f : \mathbb{R}^d \to \mathbb{R}$ is $\mu$-strongly convex if

$$f(v) \geq f(u) + \nabla f(u)^{\mathsf{T}}(v - u) + \frac{\mu}{2}\|v - u\|_2^2 \qquad \text{for all } u, v \in \mathbb{R}^d. \tag{9}$$

*Proof of Proposition 12.* First, we show that Assumption 11 implies Assumption 9. That is, we want to find a value $v_i$ for each $i$ such that all $w_i \geq v_i$ are infeasible. For some fixed $i$, define $g(w) = \epsilon(w, \mathbf{x}_i)$ and $g'(w) = \frac{\mathrm{d}}{\mathrm{d}w} g(w)$. We apply the strong convexity condition from Equation (9) to $\exp \circ g$ at $u = 1$ and $v = w$:

$$e^{g(w)} \geq e^{g(1)} + g'(1)e^{g(1)}(w - 1) + \frac{\mu}{2}\|w - 1\|_2^2$$

$$\frac{1}{w}\Big(e^{g(w)} - 1\Big) \geq \frac{\mu}{2}w + \Big(e^{g(1)} - g'(1)e^{g(1)} + \frac{\mu}{2} - 1\Big)\frac{1}{w} + g'(1)e^{g(1)} - \mu$$

$$\frac{1}{w}\Big(e^{g(w)} - 1\Big) \geq b_1 w^1 + b_0 w^0 + b_{-1} w^{-1}$$

for appropriately defined constants $b_1$, $b_0$, and $b_{-1}$. We need to distinguish two cases, based on the sign of $b_{-1}$.

**Case 1:** $b_{-1} < 0$. In this case, we have $b_{-1}w^{-1} \geq b_{-1}$ for all $w \geq 1$. A sufficient condition for infeasibility is

$$b_1 w + b_0 + b_{-1} \geq e^{\epsilon^\star} - 1$$

$$w \geq \frac{e^{\epsilon^\star} - 1 - b_0 - b_{-1}}{b_1}$$

$$w \geq \frac{e^{\epsilon^\star} - e^{g(1)} + \mu/2}{\mu/2}$$

$$w \geq 2\Big(\frac{e^{\epsilon^\star} - e^{g(1)} - \mu/2}{\mu} + 1\Big).$$

**Case 2:** $b_{-1} \geq 0$. In this case, we have $b_{-1}w^{-1} \geq 0$. Analogously to Case 1, the condition for infeasibility is

$$b_1 w + b_0 \geq e^{\epsilon^\star} - 1$$

$$w \geq \frac{e^{\epsilon^\star} - 1 - b_0}{b_1}$$

$$w \geq \frac{e^{\epsilon^\star} - g'(1)e^{g(1)} + \mu - 1}{\mu/2}$$

$$w \geq 2\Big(\frac{e^{\epsilon^\star} - g'(1)e^{g(1)} - 1}{\mu} + 1\Big).$$

We can summarize the two cases by defining

$$v_i = 2\Big(\frac{e^{\epsilon^\star} - \bar{v}_i}{\mu} + 1\Big), \quad \text{where} \quad \bar{v}_i = \min\Big\{e^{g(1)} + \frac{\mu}{2},\ g'(1)e^{g(1)} + 1\Big\}.$$

Then, $v_i$ is the desired constant for Assumption 9.

Having established that the optimal solution $w_i^\star$ is in the interval $[1, v_i]$, it remains to show that it can be found via bisection search. Bisection finds a solution to

$$\log\Big(1 + \frac{1}{w}\Big(e^{g(w)} - 1\Big)\Big) = \epsilon^\star,$$

or, equivalently,

$$e^{g(w)} = w\Big(e^{\epsilon^\star} - 1\Big) + 1. \tag{10}$$

Since the left hand-side of Equation (10) is strongly convex and the right hand-side is linear, there can be at most two solutions to the equality. We distinguish two cases.

**Case 1:** $g(1) = \epsilon^\star$ **and** $g'(1) < 1 - e^{-g(1)}$  In this case, there is a neighborhood around $w = 1$ in which we have $e^{g(w)} < w(e^{\epsilon^\star} - 1) + 1$. This implies that there are two solutions to Equation (10), one at $w = 1$ and one in $w \in (1, v_i]$. The latter is the desired solution.

**Case 2:** $g(1) < \epsilon^\star$ **or** $g'(1) \geq 1 - e^{-g(1)}$  In this case, either condition guarantees that there is exactly one solution to Equation (10) in $w \in [1, v_i]$. With the first condition, it follows from the convexity of $\exp \circ g$. If the first condition is not met but the second one is, then we have $e^{g(w)} \geq w(e^{\epsilon^\star} - 1) + 1$ for all $w \geq 1$. Then, the uniqueness follows from strong convexity.

The two cases are implemented by the if-condition in Algorithm 1. $\qquad\square$

---

**Algorithm 2** Weighted DP Lloyd's algorithm

---

**Input:** Weighted data set $\mathcal{S} = \{(w_i, \mathbf{x}_i)\}_{i=1}^n$, initial cluster centers $\mathcal{C}$, number of iterations $T$, noise scales $\beta_{\text{sum}}, \beta_{\text{count}}$
**Output:** Cluster centers $\mathbf{c}_1, \ldots, \mathbf{c}_k$
**for** $t = 1, \ldots, T$ **do**
    Compute cluster assignments $\mathcal{C}_1, \ldots, \mathcal{C}_k$, where $\mathcal{C}_j = \{\mathbf{x}_i \mid j = \arg\min_j \|\mathbf{x}_i - \mathbf{c}_j\|^2\}$
    **for** $j = 1, \ldots, k$ **do**
        Sample $\xi_j \sim \text{Lap}(0, \beta_{\text{count}})$
        Sample $\zeta_j$ from the density proportional to $\exp(-\|\zeta_j\|_p / \beta_{\text{sum}})$
        Update $\mathbf{c}_j = \frac{1}{\xi_j + \sum_{(w,\mathbf{x}) \in \mathcal{C}_j} w} \left( \zeta_j + \sum_{(w,\mathbf{x}) \in \mathcal{C}_j} w\mathbf{x} \right)$
    **end for**
**end for**

---

### D.4 Proposition 13

Algorithm 2 summarizes the weighted version of differentially private Lloyd's algorithm as introduced in Section 4.

**Proposition 13.** *The weighted DP Lloyd algorithm (Algorithm 2) satisfies the PDP profile*

$$\epsilon_{\text{Lloyd}}(w, \mathbf{x}) = \left( \frac{1}{\beta_{\text{count}}} + \frac{\|\mathbf{x}\|_p}{\beta_{\text{sum}}} \right) Tw. \tag{3}$$

*Proof.* The weighted DP-Lloyd algorithm consists of $T$ weighted sum mechanisms and $T$ weighted count mechanisms. The weighted sum mechanism is an exponential mechanism while the weighted count mechanism uses Laplace noise. We first derive the PDP profile for a general exponential mechanism and then reduce the two special cases to the general case.

Let $\mathcal{D} = \{(w_i, \mathbf{x}_i)\}_{i=1}^n \cup \{(w_0, \mathbf{x}_0)\}$ and $\mathcal{D}' = \{(w_i, \mathbf{x}_i)\}_{i=1}^n$ denote neighboring data sets. Let $\mathcal{M}(\mathcal{D})$ be a mechanism taking values in $\mathbb{R}^d$, distributed according to the probability density function $f(y) \propto \exp(-k\, d(y, g(D)))$ where $k > 0$, $d(\cdot, \cdot)$ is a metric on $\mathbb{R}^d$ and $g : [1, \infty) \times \mathcal{X} \to \mathcal{Y}$ is a function where $\mathcal{Y} \subseteq \mathbb{R}^d$. In order to bound the privacy loss of $\mathcal{M}$, we require that the influence on $g$ of any single weighted point $(w_0, \mathbf{x}_0)$ be bounded. Specifically, we require that there exist a function $\Delta : \mathcal{X} \to \mathbb{R}_{\geq 0}$ such that

$$d(g(\mathcal{D}), g(\mathcal{D}')) \leq w_0 \Delta(\mathbf{x}_0) \quad \text{for all neighboring } \mathcal{D}, \mathcal{D}' \text{ differing in } (w_0, \mathbf{x}_0). \tag{11}$$

Given these prerequisites, we can see that the PDP profile of $\mathcal{M}$ is bounded as

$$
\begin{aligned}
\log\left|\frac{f(y)}{f'(y)}\right| &= \left|-k\left(d(y, g(\mathcal{D})) - d(y, g(\mathcal{D}'))\right) + \log\frac{\int_{\mathbb{R}^d}\exp(-k\,d(y, g(\mathcal{D})))\,\mathrm{d}y}{\int_{\mathbb{R}^d}\exp(-k\,d(y, g(\mathcal{D}')))\,\mathrm{d}y}\right| \\
&\leq k\,|d(y, g(\mathcal{D})) - d(y, g(\mathcal{D}'))| \\
&\leq k\,d(g(\mathcal{D}), g(\mathcal{D}')) \\
&\leq k\,w_0\Delta(\mathbf{x}_0).
\end{aligned}
$$

The first inequality follows because the two integrals are identical due to translation invariance (note that the integrals are over all of $\mathbb{R}^d$). The second inequality follows from the triangle inequality for the metric $d$. The third inequality follows from Equation (11).

Having established the general case, we apply it to the two submechanisms that constitute the weighted DP-Lloyd algorithm.

- The weighted sum mechanism is an exponential mechanism for $g(D) = \sum_{(w,\mathbf{x})\in D} w\,\mathbf{x}$ and $d(y, y') = \|y - y'\|_p$ for some $p > 0$ and $k = 1/\beta_{\mathrm{sum}}$. The sensitivity function is $\Delta(\mathbf{x}) = \|\mathbf{x}\|_p$ because

$$
d(g(D), g(D')) = \|w_0\,\mathbf{x}_0\|_p = w_0\,\|\mathbf{x}_0\|_p.
$$

  Thus, $|\log f(y)/f'(y)| \leq w_0\|\mathbf{x}_0\|_p/\beta_{\mathrm{sum}}$.

- The weighted count mechanism is an exponential mechanism for $g(D) = \sum_{(w,\mathbf{x})\in D} w$ and $d(y, y') = |y - y'|$ and $k = 1/\beta_{\mathrm{count}}$. The sensitivity function is $\Delta(\mathbf{x}) = 1$ because

$$
d(g(D), g(D')) = |w_0| = w_0.
$$

  Thus, $|\log f(y)/f'(y)| \leq w_0/\beta_{\mathrm{count}}$.

The result now follows by adaptive composition over the $T$ iterations of the algorithm. $\square$

