# OpenReview forum: "Personalized Privacy Amplification via Importance Sampling"
_TMLR — Accepted by TMLR_

### Review · Reviewer_dK8Z · 2024-10-03

**Summary Of Contributions:**

This work proposes a novel framework for importance sampling in the context of personalised differential privacy. Here the privacy-utility-efficiency tradeoff is explored and two approaches are proposed: one based on the privacy-optimal sampling and one based on the concept of coresets. The work demonstrates the results both theoretically and empirically (k-means clustering) on a number of datasets.

**Audience:**

Yes

**Broader Impact Concerns:**

No specific concerns.

**Claims And Evidence:**

Yes

**Requested Changes:**

For k-means there are no results with epsilons of 0 and infinity, meaning that the other ones are more difficult to contextualise, I would encourage the authors to add these.

Text-based changes:

There is not enough explanation for how to interpret figure 3, it could benefit from a simplified summary.

Explain the interplay between utility and efficiency in a broader setting.

Add concrete explanation for the broader context of the work's contributions.

Your figures 3 and 4 are difficult to interpret for people with red-green colorblindness. You should change these.

Overall (given the broad community of TMLR), I believe this work is of interest and of relevance. I have checked the majority of the theorems, but have not gone into the proofs in the appendix in more detail, so these seem sound to me.

So in general, provided the authors can respond to questions and concerns well and no other fundamental issues are raised by the reviewers, I would lean towards acceptance.

**Strengths And Weaknesses:**

Strengths:

The work is well written and well-motivated. The methods are (to the best of my knowledge) sound (conditional on some clarifications I requested below) and are definitely of interest to the TMLR readers as well as a privacy research community. I particularly like the split of the privacy-utility into a privacy-utility-efficiency discussion, making this more of a relevant discussion to settings such as distributed ML.

Weaknesses:

The concept of poisson sampling for privacy amplification is hardly new and has been thoroughly explored in prior works and dp frameworks (e.g. dp-transformers or opacus to name a few), I am not sure exactly what is the reader meant to gain as the main message of the work? That you can also have importance sampling with poisson which is more efficient? Or that it produces a better PDP profile? The actual value of the contributions is a bit lost on me here.

Further to my comment on efficiency-utility: I am actually struggling to see the practical difference between the terms (i.e. higher efficiency means that the utility of a ML model is higher and vice versa) and am not convinced that this part of the manuscript explains the interplay well/correctly. Particularly with respect to samples which have a higher privacy loss (as these are likely to be 'memorised' by the model) and samples of high utility are often those that are both important and useful to the model. Can you elaborate on this, because I do not see straightaway what the trade-off is here between utility and efficiency? And, hence, I would have also expected them to have an identical/similar privacy trade-off too.

This is more of a clarification than a weakness: if one starts weighting the selection of each sample differently, this would mean that the actual privacy loss of individuals in the dataset would be very different (and it may even occur that some of them go over the privacy budget), how is this accounted for in this framework? Or am I missing something here?

Questions:

Is there a principled way to reason over m? In some of your experiments its arbitrarily set to 19, in others searched over a list. Can this quantity be chosen (somewhat) reasonably without the extensive knowledge of the underlying dataset or obtaining data-related statistics?

I assume that this is only true for real independent variables and settings (i.e. the sampling is unlikely to have meaningful interpretation in deep learning models or in time-series datasets, for instance?)

When discussing the epsilon level of k-means from a 2-step process, how are the budgets composed across them?

Are the reported privacy losses expected to be very different across samples?

I did not fully understand this phrase: 'Notably, with privacy-optimal sampling, we can achieve some sampling “for free” by assigning a
sampling probability of 1 to the data point(s) with the highest privacy loss and probabilities less
than 1 to the remaining points.' Could you elaborate on why exactly is this the case?

---

> ### Author Response · Authors · 2024-10-30
> **Response to Reviewer dK8Z (1)**
>
> Thank you very much for your thorough review!
> We find your suggested changes helpful and have updated our manuscript to reflect them.
> We will upload the updated version by the end of the week.
> Below, we respond to your individual comments and questions.
>
> > **Weakness**: The concept of poisson sampling for privacy amplification is hardly new [...] I am not sure exactly what is the reader meant to gain as the main message of the work? [...] The actual value of the contributions is a bit lost on me here.
> > **Requested Change:** Add concrete explanation for the broader context of the work's contributions.
>
> While Poisson sampling has been used since the beginnings of differential privacy, it has been studied predominantly with uniform sampling probabilities and without an individualized privacy analysis.
> Recent work [[1]](https://arxiv.org/abs/2007.12674) [[2]](https://arxiv.org/abs/2408.07006) considered data-dependent sampling for differential privacy, but found utility to be *at odds* with privacy because of an additional privacy leakage due to the data-dependence in the sampling.
> In contrast, we demonstrate that privacy and utility are actually *aligned* goals when Poisson importance sampling (which we introduce in Definition 5) is used.
> Thus, our high-level message is that importance sampling is a more promising tool for differential privacy than previously thought.
> Specifically, it allows us to achieve a better privacy-utility trade-off than uniform sampling, at any given sample size.
>
> > **Weakness**:  I am actually struggling to see the practical difference between [utility and efficiency]
> > **Requested Change:** Explain the interplay between utility and efficiency in a broader setting.
>
> Thank you for pointing this out.
> We realized that our introduction used the term *efficiency* to refer both to *computational efficiency* and to *statistical efficiency* without disambiguating properly.
> In our updated version, we have eliminated all uses of *statistical efficiency* to avoid confusion.
> Thus, the efficiency-utility trade-off simply refers to the fact that we can train a more accurate model the more compute we have available, as you pointed out.
> Specifically, in the context of subsampling, the primary indicator for efficiency is the subsample size: the lower, the less compute is required by the subsequent algorithm.
>
> > **Requested Change:** Your figures 3 and 4 are difficult to interpret for people with red-green colorblindness. You should change these.
>
> Thank you for highlighting this.
> We use the colorblind-friendly palette suggested by [Wong (2011)](https://www.nature.com/articles/nmeth.1618).
> In the updated manuscript, we have picked a different selection from the palette and added texture to the shaded areas to improve visibility further.
>
> > **Requested Change:** For $k$-means there are no results with epsilons of 0 and infinity, meaning that the other ones are more difficult to contextualise, I would encourage the authors to add these.
>
> We revised Figures 3 and 6 and added two horizontal reference lines: the performance of uniform subsampling and coreset-based subsampling for the case of $\epsilon=\infty$. Both reference lines seem to overlap, however, note that the line for coreset-based is lower than the line for uniform. For our privacy-optimal sampling strategy there is no non-private counterpart (the resulting sample size would be zero).
>
> In Figures 4 and 7, we also added a reference line: the black line depicts the performance of DP-Lloyd with the same privacy parameter but on the full data set. Once again, we see that all sampling strategies approach the black line as the subsample size $m$ increases. However, our proposed sampling strategies are faster than uniform in doing so.
>
> We are unsure what you mean by $\epsilon=0$, however.
> Note that, under $\epsilon=0$, no information can be processed at all, which prevents us from doing any meaningful comparison.
> If there is additional context you would like us to add, could you please clarify?

---

> ### Author Response · Authors · 2024-10-30
> **Response to Reviewer dK8Z (2)**
>
> > **Question/Weakness:** if one starts weighting the selection of each sample differently, this would mean that the actual privacy loss of individuals in the dataset would be very different (and it may even occur that some of them go over the privacy budget), how is this accounted for in this framework?
> > **Question:** Are the reported privacy losses expected to be very different across samples?
>
> The key is that the base mechanism (without sampling or weighting) already has a heterogeneous privacy profile, see Figure 1 (color encodes individual privacy loss before sampling).
> Privacy-optimal sampling actually acts as an equalizer: after sampling, each point has exactly the same individual privacy loss due to constraint (2b).
> In coreset-based sampling, the individual privacy losses are heterogeneous but bounded -- see our discussion surrounding Eq. (5).
> Moreover, Figure 1 shows that the coreset-based weights are clearly correlated to the privacy-optimal weights, so we should expect coreset-based sampling to also have a flattening effect.
>
> > **Question:** Is there a principled way to reason over m?
>
> Increasing $m$ improves accuracy and privacy, but decreases efficiency.
> In a compute-constrained environment, we would always choose $m$ as large as possible subject to the compute constraints we operate under.
> Note that coreset-based sampling is only well-defined for $m \leq n \tilde{x}/r$ where $\tilde{x}$ is the average squared $\ell_2$ norm.
> For larger values of $m$, privacy-optimal sampling should be used.
>
> > **Question:** When discussing the epsilon level of k-means from a 2-step process, how are the budgets composed across them?
>
> We use basic (i.e., additive) composition.
>
>
> > **Question:** I did not fully understand this phrase: 'Notably, with privacy-optimal sampling, we can achieve some sampling “for free” [...]'
>
> We agree that there was not enough context in the introduction for this statement.
> In the updated manuscript, we have moved this to Section 3 and elaborated how we can have sampling for free as follows:
> > Notably, with privacy-constrained sampling, we can achieve some sampling "for free": if the base mechanism satisfies $\epsilon$-DP and we choose $\epsilon^*=\epsilon$, then the sampled mechanism also satisfies $\epsilon$-DP while operating on a smaller sample.

---

> > ### Comment · Reviewer_dK8Z · 2024-10-30
> > **Response to the rebuttal**
> >
> > I would like to thank the authors for a comprehensive response. My concerns have been addressed (provided these are now reflected in the text of the manuscript).
> >
> > When talking about the epsilon of 0, I meant the baseline model (i.e. no training). Now given it a second thought, it might not be super relevant to the figure itself, but it should just be mentioned in the caption/text somewhere (to contextualise that DP can bring a lot of utility even in strictest privacy settings when compared to using algorithms off-the-shelf).
> >
> > Provided the text changes outlined above are made to the manuscript, I would recommend acceptance. I would like to see an updated manuscript (if possible) and (again, if possible) a diff file.

---

> > > ### Author Response · Authors · 2024-10-31
> > > **Response to baseline clarification**
> > >
> > > Thank you for the clarification. We have added the following text:
> > >
> > > > Note also that, as $\epsilon \to 0$, the cost approaches but stays below that of performing no learning at all.
> > > This can be seen by comparison to the $\tilde{x}$ column of Table 1: when the cluster centers are initialized to zero, the cost is precisely $\tilde{x}$, i.e., the average squared $\ell_2$ norm.

---

### Review · Reviewer_HJwH · 2024-10-12

**Summary Of Contributions:**

This article studies the relationship b/w privacy, utility, and importance (or informativeness of samples) to understand how important sampling may help improve privacy guarantees. The primary insight by the authors is that leveraging the most informative samples (which generally have high individual privacy loss) will not only lead to better utility (expected) but rather will also provide better privacy protection and thus these samples need to be chosen more often. Directly and counterintuitively, this implies that a smaller expected number of samples is often preferred (the most informative samples) to achieve good utility and privacy.

After setting up an appropriate optimization problem for minimizing the expected sample size (by optimizing for the sampling weights) over the individual privacy profiles, the authors provide an algorithm for achieving the same. Furthermore, empirical results for the k-means problem verify the solution proposed. Overall, this work provides valuable insights into the synergy between individual privacy and importance sampling

**Audience:**

Yes

**Claims And Evidence:**

Yes

**Requested Changes:**

The changes requested are mainly on the presentation of the sample size results. Please refer to the weaknesses section and provide some more details on how the sample size is different compared to traditional sampling (mainly on population size, and privacy when the same sample is chosen more often than others).

**Strengths And Weaknesses:**

Strengths:

1. The work provides interesting insights into how a higher expected sample size may worsen overall privacy loss when individual privacy losses are known.
2. Similarly, the synergy b/w informativeness and high-privacy loss is a valuable tool that is intuitive but it was useful to have actual numerical examples to verify the same.
3. Designing individual sampling weights with privacy in mind is a broad enough problem that deserves attention in several learning techniques and can be quite valuable.
4. Overall presentation and empirical results look good.

Weaknesses:
1. From Algorithm 1, it's unclear how the privacy of the weights is maintained (since they depend on the training dataset/dataset's privacy which may leak some information about the data). Adding exact details here would be useful
2. From the optimization problem, it makes sense that having a lower expected sample size is good based on the sampling weights. But there seems to be some confusion here. a) I think a larger population size is still desired, and I believe in the current writeup it seems that we may want the total sample size to be small. Adding some clarification here will be good. b) From a hypothetical point of view, how does problem 7 handle the following case? Suppose, a single point is significantly more informative than others (and the final probabilities are 0.99 for the one point, v/s 1e-3 for the rest) then will the same point get chosen each time with high probability? (this is probably more applicable to something like an SGD problem v/s k-means).
3. What is the applicability of the problem setting in 7 to problems other than K-means? It would be nice to get some insights here.
4. For the user-DP or local-DP scenario, how can algorithm 1 be modified? Wherein the privacy costs for each sample may be unknown or need to be disclosed under DP as well.
5. Curious, if without privacy guardrails, only the most informative samples are chosen then can we observe better utility? This could be similar to the coreset setting.

---

> ### Author Response · Authors · 2024-10-30
> **Response to Reviewer HJwH**
>
> Thank you very much for your thorough review!
> We find your suggested changes helpful and have updated our manuscript to reflect them.
> We will upload the updated version by the end of the week.
> Below, we respond to your individual comments and questions.
>
> > From Algorithm 1, it's unclear how the privacy of the weights is maintained (since they depend on the training dataset/dataset's privacy which may leak some information about the data). Adding exact details here would be useful
>
> Thank you, we agree that more specifics would be useful.
> In the updated manuscript, we elaborate on the privacy of the weights in general in Section 3.1 (after Definition 5); and for privacy-optimal sampling in Section 3.2 (after Theorem 10).
> Essentially, the weights are an intermediate result and are not published.
> Their impact on the privacy loss is accounted for in the proof of Theorem 6.
>
> > I think a larger population size is still desired, and I believe in the current writeup it seems that we may want the total sample size to be small. Adding some clarification here will be good.
>
> We face a trade-off between efficiency on one side, and privacy and utility on the other.
> A small sample size is good for efficiency but bad for privacy and utility.
> We have made clarifications regarding the involved trade-offs in the introduction.
>
>
> > From a hypothetical point of view, how does problem 7 handle the following case? [...]
>
> First, recall that we are using Poisson sampling, so each point can only be selected once.
> Next, if one point has a high selection probablity and all others are low, this may indicate that the one point is an outlier.
> Since we bound user contributions at a quantile, this is not a concern in our set-up.
> Finally, in the context of SGD, it would be unusual to observe this throughout multiple iterations because previusly sampled points should have smaller gradients in subsequent iterations.
> If you have a specific problem in mind, we would be happy to comment in more detail given the specifics.
>
> > What is the applicability of the problem setting in 7 to problems other than K-means? It would be nice to get some insights here.
>
> Problem 7 is not specific to $k$-means.
> It is applicable to any $\epsilon$-DP mechanisms with known PDP profile.
>
> > For the user-DP or local-DP scenario, how can algorithm 1 be modified? Wherein the privacy costs for each sample may be unknown or need to be disclosed under DP as well.
>
> As far as we are aware, Algorithm 1 does not introduce any limitations to heterogeneous privacy requirements, because individualized privacy parameters do not affect the separability of the problem.
> In particular, if the individual privacy parameters are public, Algorithm 1 can be used directly by replacing $\epsilon^*$ in constraint (2b) with the respective individual parameter.
>
> > Curious, if without privacy guardrails, only the most informative samples are chosen then can we observe better utility? This could be similar to the coreset setting.
>
> Yes, the idea of a coreset is precisely to choose informative data points.
> Without the privacy guardrails, we fall back to coresets and those perform better than uniform subsamples, see [[1]](https://dl.acm.org/doi/10.1145/3219819.3219973).

---

> > ### Author Response · Authors · 2024-11-21
> > **Follow-up on response**
> >
> > Dear reviewer,
> >
> > Does the updated manuscript address your concerns? Please let us know if any questions remain open.

---

### Review · Reviewer_5tAQ · 2024-10-16

**Summary Of Contributions:**

This paper studies the problem of importance sampling within the framework $\epsilon$-PDP (Personalized differential privacy).
Specifically, this paper proposed a subsampling algorithm towards optimizing the expected sampled size (mimicking the efficiency objective) for a given PDP algorithm $M$, while bounding the subsampled algorithm $M\circ S_q$ to maintain a privacy guarantee under $\epsilon^⋆-$DP.

Furthermore, the paper explores the application of importance sampling to the differentially private k-means clustering problem and introduce a coreset-based sampling scheme.

Experimental results demonstrate that importance sampling can generally improve the utility-/efficiency- privacy trade-off compared to uniform sampling.

**Audience:**

Yes

**Broader Impact Concerns:**

Since this paper focuses on a rigorous privacy guarantee framework and the experiments are conducted on standard public datasets,  I do not identify any ethical concerns.

**Claims And Evidence:**

Yes

**Requested Changes:**

- The objective defined in Eq. (2a) would be more accurately described as "efficiency-optimal" rather than "privacy-optimal" sampling.

- The experimental setup in Section 3.3 could be clarified further. It would be helpful to explicitly state the $\epsilon(w_i, x_i)$  function used in the experiment. Additionally, providing the formula for the MSE visualized in Figure 2 would improve clarity.

- In Figure 2 (right), the MSE achieved by variance-optimal sampling appears to outperform the proposed “privacy-optimal” approach. It would be insightful to explain the rationale for not always using variance-optimal sampling and to highlight the principled advantages of the proposed importance sampling method.

- Page 22, Equation (6): It seems that the multipliers for the first and second terms may need to be $q(x)$ and $(1-q(x))$ instead of $(1-q(x))$ and $q(x)$.

- Page 23, Theorem 10:
The statement: “Assumption 8 guarantees that the feasible region is bounded, and Assumption 9 guarantees that it is non-empty. Since the objective is strictly monotonic, the solution must be unique.”
should be corrected to:
“Assumption 8 ensures the feasible region is non-empty, and Assumption 9 guarantees it is bounded.”


- Page 24, Proof of Proposition 12: For consistency, the notation $\mu_i$​ should be changed to $\mu$.

**Strengths And Weaknesses:**

**Strengths:**
- The use of importance sampling for DP mechanisms addresses a significant problem.
- The paper explores a natural application of importance sampling in DP, i.e., the DP k-means clustering.
- The experimental results provide promising evidence, demonstrating the benefits of using importance sampling over trivial uniform sampling in relevant scenarios.


**Weaknesses:**
- The study is largely confined to the $\epsilon$-PDP framework, limiting its applicability to other widely adopted notions of differential privacy (including the technically more interesting ones). Moreover, given that the work builds upon the $\epsilon$-PDP framework, some of the subsequent derivations appear relatively straightforward.
- The overall presentation could benefit from improvements. Stronger logical connections between key points, more intuitive motivation to enhance understanding, and clearer descriptions of the experimental results would significantly improve the paper. Additionally, several confusing sections are noted below that could be clarified further.

---

> ### Author Response · Authors · 2024-10-31
> **Response to Reviewer 5tAQ**
>
> Thank you for your detailed comments.
> We find your suggested changes helpful and have updated our manuscript to reflect them.
> We will upload the updated version by the end of the week.
> Below, we respond to your individual comments and questions.
>
> > The study is largely confined to the $\epsilon$-PDP framework, limiting its applicability to other widely adopted notions of differential privacy (including the technically more interesting ones). Moreover, given that the work builds upon the $\epsilon$-PDP framework, some of the subsequent derivations appear relatively straightforward.
>
> While it is true that relaxed versions of DP are often preferred in practice, we believe that our results will nevertheless be of interest to the DP and coreset communities.
> Part of our contribution lies in identifying a setting in which utility and privacy are well aligned and which can be used constructively for algorithm design.
> This part is not obvious which is evidenced by the recent negative results on non-uniform sampling [[1](https://arxiv.org/pdf/2007.12674), [2](https://arxiv.org/pdf/2408.07006)] as well as by the absence of importance sampling techniques from the DP machine learning literature, despite their prevalence in the coreset literature.
> In this context, we do not agree that simplicity should be seen as a weakness.
>
> > The overall presentation could benefit from improvements. Stronger logical connections between key points, more intuitive motivation to enhance understanding, and clearer descriptions of the experimental results would significantly improve the paper.
>
> We have clarified the manuscript in several places, including the general context of the work and the description of the experiments.
> We hope you find these adjustments helpful.
> If you have additional remarks on the presentation, we would be happy to address them.
>
> > **Requested Change:** The objective defined in Eq. (2a) would be more accurately described as "efficiency-optimal" rather than "privacy-optimal" sampling.
>
> Thank you for raising this point. We rephrased it to *privacy-constrained*.
> We prefer this term because we would like to highlight that it is informed by a privacy criterion in contrast to coreset-based sampling which is utility-informed.
>
> > **Requested Change:** The experimental setup in Section 3.3 could be clarified further. It would be helpful to explicitly state the $\epsilon(w_i, x_i)$ function used in the experiment. Additionally, providing the formula for the MSE visualized in Figure 2 would improve clarity.
>
> Thank you for the suggestion.
> In the updated manuscript, we now explicitly state the PDP profile and the quantities between which the MSE is computed.
>
> > **Requested Change:** In Figure 2 (right), the MSE achieved by variance-optimal sampling appears to outperform the proposed “privacy-optimal” approach. It would be insightful to explain the rationale for not always using variance-optimal sampling and to highlight the principled advantages of the proposed importance sampling method.
>
> That is a good point, we have clarified this in the updated manuscript.
> The variance-optimal sampling is an idealized benchmark that does not satisfy $\epsilon$-DP.
> It would be the optimal choice under relaxed conditions, so it serves as a lower bound on the achievable variance.
>
> > **Requested Change:** Page 22, Equation (6): It seems that the multipliers for the first and second terms may need to be $q(x)$ and $(1-q(x))$ instead of $(1-q(x))$ and $q(x)$.
>
> > **Requested Change:** Page 23, Theorem 10: The statement: “Assumption 8 guarantees that the feasible region is bounded, and Assumption 9 guarantees that it is non-empty. Since the objective is strictly monotonic, the solution must be unique.” should be corrected to: “Assumption 8 ensures the feasible region is non-empty, and Assumption 9 guarantees it is bounded.”
>
> > **Requested Change:** Page 24, Proof of Proposition 12: For consistency, the notation $\mu_i$ should be changed to $\mu$.
>
> Thank you very much for your careful reading! You are correct and we have implemented the changes in the updated manuscript.

---

> > ### Author Response · Authors · 2024-11-21
> > **Follow-up on response**
> >
> > Dear reviewer,
> >
> > Does the updated manuscript address your concerns? Please let us know if any questions remain open.

---

### Author Response · Authors · 2024-11-04
**Revision and response to all reviewers**

We thank the reviewers for their thoughtful comments.
We have made a revision that addresses the reviewers' concerns.
**All changes are highlighted in blue.**
The requested changes were primarily clarifications/elaborations, improvements to the presentation of the results and minor corrections in the proofs.
Specifically, we have made the following changes, as stated in the revision notes:
- Clarified the trade-offs between privacy, utility and efficiency (**dK8Z**, **HJwH**)
- Clarified the nature of the variance-optimal distribution (**5tAQ**)
- Clarified the privacy of Algorithm 1 (**dK8Z**, **HJwH**)
- Clarified the experimental set-up (**dK8Z**, **5tAQ**)
- Additional reference points for the experimental results (**dK8Z**)
- Improved visibility of the plots (**dK8Z**)
- Renamed *privacy-optimal* to *privacy-constrained* (**5tAQ**)
- Minor error corrections in the proofs (**5tAQ**)

We would also like to note that all reviewers agree that the work is of interest to the TMLR and privacy communities.
Further, the reviewers chose to highlight that the work is well presented (**dK8Z**, **HJwH**); that the discussions of the various privacy-utility-efficiency trade-offs is interesting (**dK8Z**, **HJwH**); and that the empirical results are promising (**HJwH**, **5tAQ**).

---

### Decision · Action_Editor_LxfE · 2024-12-02

**Recommendation:** Accept as is

**Comment:**

The reviewers unanimously agreed that the paper is worth of publication so the decision is clear. I also think the results of Section 3, including the optimization based approach of Section 3.2, are very interesting.
Comment: you mention in the conclusions that "Promising directions for future work include extensions to $(\varepsilon,\delta)$-DP or R\'enyi-DP...", however no details on possible RDP aspects are givne. Perhaps the RDP-filtering analysis by Feldman and Zrnic could be quite directly used to get similar results for RDP? If you think so, feel free to add a sentence or two, though this is not necessary.

**Audience:**

The primary audience for this paper is researchers working in DP and privacy-preserving machine learning. It is also quite easily approachable and probably also quite interesting to people without a lot of background in DP.

**Claims And Evidence:**

The paper proposes a framework for importance sampling within the Personalized Differential Privacy (PDP) setting, aiming to enhance the trade-offs between privacy, utility, and efficiency. The starting point is the observation that the most informative samples can improve both privacy and utility while reducing the expected sample size compared to uniform sampling. This is also in line with the experimental observations made by [Yu et al., TMLR 2023](https://arxiv.org/pdf/2206.02617). The claims are supported by theoretical results, including an optimization problem that balances individual privacy losses with informativeness, and empirical results demonstrating improved performance in private $k$-means clustering. However, the approach is largely confined to the PDP framework, limiting its broader applicability in its current form.